# Meta-optics redefines microdisplay: monolithic color LCoS without polarization dependency

Xiangnian Ou[1,6], Yueqiang Hu [1,2,3,6] ✉, Dian Yu[1], Shulin Liu[1], Shaozhen Lou[1], Zhiwen Shu[2], Wenzhi Wei[1], Man Liu[1], Jianxiong Li[4], Tianhai Chang[4], Na Liu [5] ✉ & Huigao Duan [1,2] ✉

Liquid crystal on silicon (LCoS) panels are pivotal to high-resolution optical projection and imaging displays, yet their inherent polarization sensitivity and reliance on multi-chip architectures for color reproduction constrain the upper limit of light utilization, increase system complexity and restrict broader applicability. Here, we demonstrate a monolithic color meta-LCoS prototype that integrates dual-layer metasurfaces to achieve polarization-insensitive, full-color amplitude modulation on a single chip. Polarization sensitivity is eliminated via a synergistic design combining metasurface-enabled polarization conversion and voltage-controlled liquid crystal phase modulation, achieving a high-contrast, polarization-insensitive optical switch. By embedding red, green, and blue metasurface subpixels and meticulously designed off-axis angles, enabling direct color synthesis through a unified device. We showcase a 64-pixel monochrome and a 9-pixel color prototype capable of dynamically projecting diverse patterns under unpolarized illumination. Fully compatible with existing LCoS fabrication processes, our device significantly reduces system complexity and cost, offering transformative applications in next-generation projectors and AR/VR displays.

Microdisplay optical engine, as core components of augmented reality (AR), virtual reality (VR), head-up displays (HUDs), and pico-projectors, demand ultra-compact size, high resolution, and high optical efficiency to meet the requirements of next-generation near-eye and projection systems[1]. These technologies are primarily based on microdisplay panels such as digital micromirror device (DMD), liquid crystal on silicon (LCoS), micro-organic light-emitting diodes (micro-OLED), and micro-light-emitting diodes (micro-LED)[2,3]. Non-emissive DMD and LCoS panels offer superior brightness, higher maturity, and greater optical design flexibility compared to emissive micro-OLED and micro-LED, making them still dominant in projection and imaging

displays. DMD, a MEMS-based reflective display composed of millions of tilting micro-mirrors (Fig. 1a), modulates light through high-speed binary switching, enabling high contrast and brightness without the need for polarized illumination[4]. However, DMD's reliance on mechanical actuation imposes limitations on resolution (pixel sizes >5 μm) and long-term reliability, while its use of a color wheel for sequential color projection compromises both color accuracy and efficiency.

LCoS, by contrast, is a reflective microdisplay technology that integrates liquid crystal (LC) modulation with silicon backplane circuitry. With its high pixel density and fill factor, LCoS surpasses DMD in

[1]National Research Center for High-Efficiency Grinding, College of Mechanical and Vehicle Engineering, Hunan University, Changsha, P.R. China. [2]Greater Bay Area Institute for Innovation, Hunan University, Guangzhou, P.R. China. [3]Advanced Manufacturing Laboratory of Micro-Nano Optical Devices, Shenzhen Research Institute, Hunan University, Shenzhen, P.R. China. [4]Huawei Technologies Co., Ltd., Bantian, Longgang District, Shenzhen, P.R. China. [5]2nd Physics Institute, University of Stuttgart, Stuttgart, Germany. [6]These authors contributed equally: Xiangnian Ou, Yueqiang Hu. ✉e-mail: huyq@hnu.edu.cn; na.liu@pi2.uni-stuttgart.de; duanhg@hnu.edu.cn

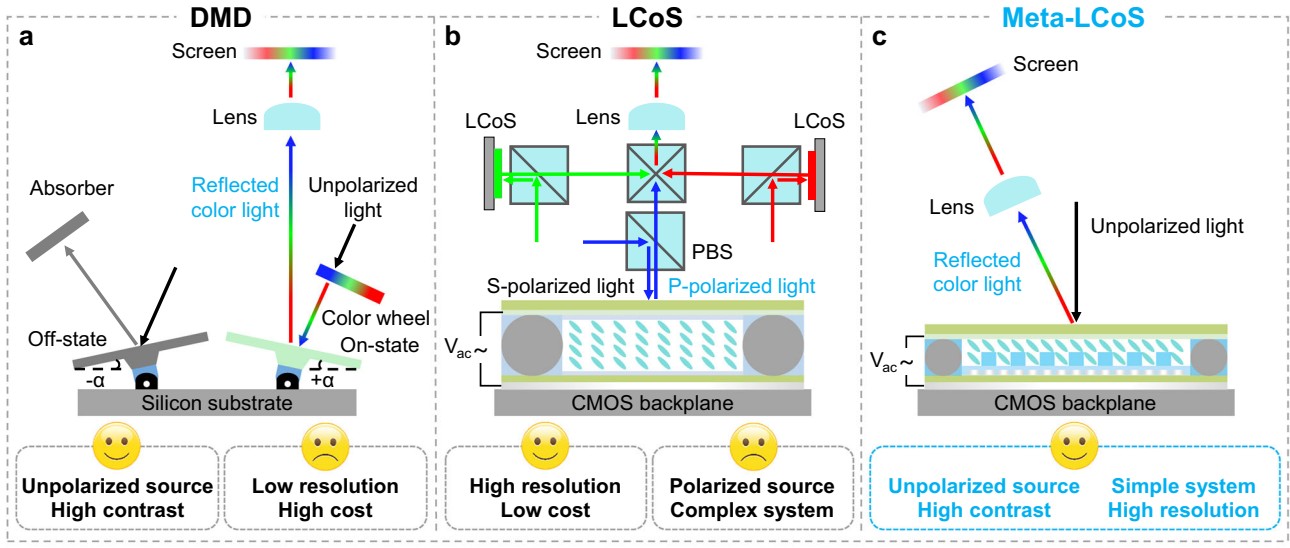

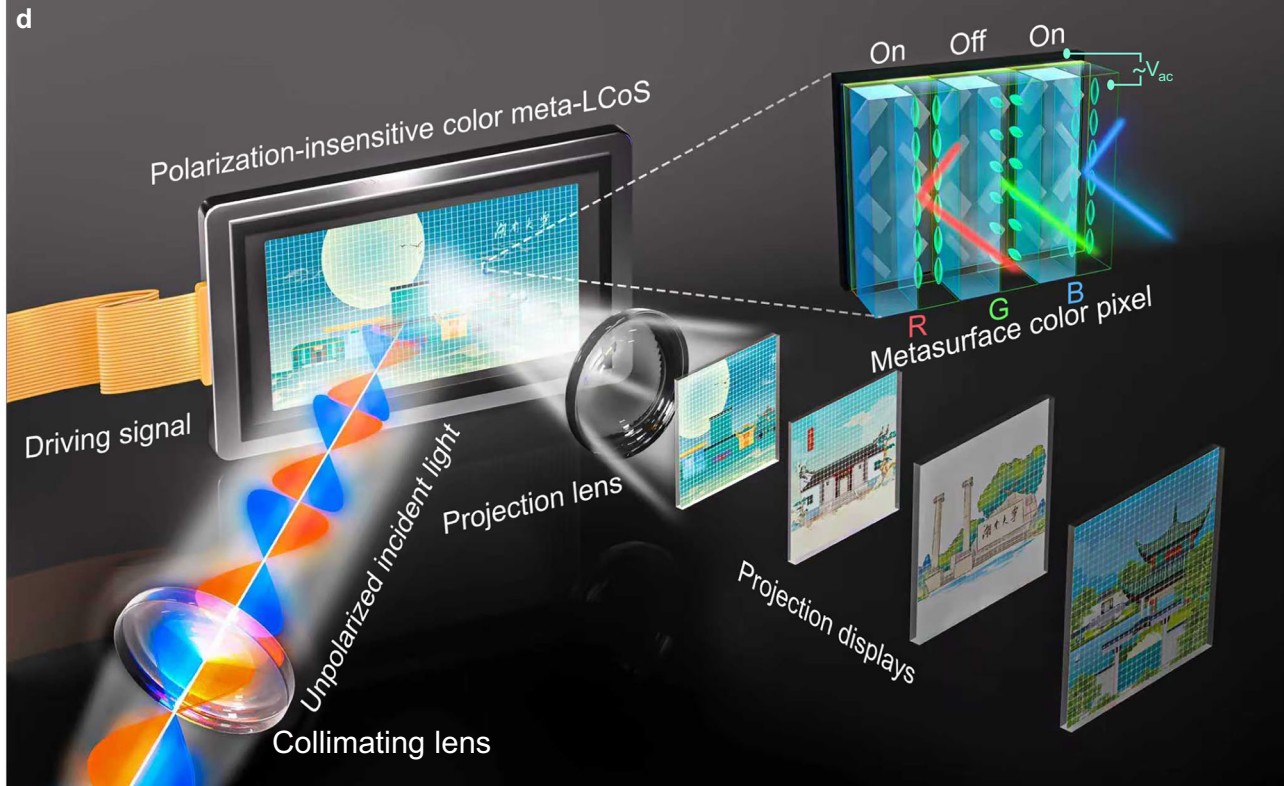

**Fig. 1 | Schematics of the display principle of the optical engines and the proposed metasurface-integrated polarization-insensitive color LCoS microdisplay. a–c** Schematic diagrams of the optical engines for color projection display: DMD (**a**), LCoS (**b**) and the proposed polarization-insensitive color meta-LCoS (**c**). The lens serves as the projection lens for focusing and magnifying the images. PBS is the polarizing beam splitter cube. With meticulous design, the proposed meta-LCoS enables high-contrast full-color amplitude modulation under unpolarized illumination, eliminating the need for bulky beam splitters and combiners and thereby simplifying the projection system. **d** Schematic of the proposed polarization-insensitive color meta-LCoS projection display. The programmable color images, driven by different digital signals, can be dynamically displayed in the far field under unpolarized incident light with a monolithic meta-LCoS. Original architectural images adapted with permission from Huitu.com.

resolution scalability, making it particularly suited for ultra-high-definition applications, including 8 K and beyond[5]. However, LCoS inherently relies on linearly polarized light, necessitating bulky polarizing optics (Fig. 1b) and limiting light utilization efficiency to ≤50% when using unpolarized sources like LEDs. Additionally, the linear polarization output limits adaptability in applications like HUDs, where polarized glasses can block the projected images[6]. It also struggles to adapt to multi-polarization scenarios, such as orthogonal polarization 3D displays[7] or multi-focus systems with varying polarization states[8,9]. Two strategies have been proposed to address these challenges: using polarization-insensitive LC materials[10,11], which require higher drive voltages and advanced packaging, or modifying the LCoS structure by integrating thin-film waveplates[12], which increases drive voltage due to added material thickness. Another fundamental limitation of LCoS and reflective microdisplays in general is the challenge of achieving monolithic full-color integration. Conventional LCoS-based color

projection systems typically employ a three-chip architecture with beam splitters and combiners, significantly increasing system cost and optical alignment complexity. While sequential color projection using a time-multiplexed LED or laser source is an alternative, this approach compromises temporal resolution and dynamic range[13].

In this work, we propose and experimentally demonstrate a polarization-insensitive monolithic color metasurface-integrated LCoS (meta-LCoS) prototype for color projection displays as illustrated in Fig. 1c. Metasurfaces, which are two-dimensional artificial materials arrayed with subwavelength-sized scatterers, have emerged as a transformative platform for light modulation[14–19]. Previously, the introduction of various mechanisms to impart tunability to metasurfaces for realizing novel multifunctional dynamic devices has garnered widespread interest[20–25]. However, there has been little consideration of how metasurfaces can enhance the capabilities of mature optical modulation devices[26]. While conventional polarization-insensitive LC or thin-film waveplates suffer from fixed optical constraints dictated by their material composition, metasurfaces achieve tunable electromagnetic responses via nanoscale structural design, offering superior adaptability for LCoS panel. Here, the metasurfaces are integrated onto the aluminum (Al) electrode pixels of the LCoS panel to eliminate polarization-sensitive properties while achieving a high-contrast optical switch, as conceptually shown in Fig. 1d. Each color pixel consists of three metasurface super unit cells dedicated to red, green, and blue, each featuring paired nanorods overlaid with alternating subwavelength nanograting and LC. The combination of polarization conversion in the metasurface and polarization-sensitive phase modulation in the LC allows the device to function with unpolarized light. An initial π-phase difference between the two columns creates destructive interference for the "off" state, while voltage control of the LC modulates the phase difference to enable high-contrast switching between "on" and "off". By pixelating the two-dimensional electrodes, a 64-pixel monochrome and a 9-pixel color 2D addressable prototype chip is achieved, enabling the dynamic generation and projection of programmable intensity images. Fully compatible with existing LCoS fabrication processes, this metasurface-integrated design reduces system complexity, enhances energy efficiency, and lowers production costs. Our findings open new avenues for next-generation projection and near-eye display technologies, offering promising applications in AR/VR, pico-projectors, and HUD systems.

## Results

### Mechanism and design of the polarization-insensitive optical switching

Figure 2a illustrates the detailed structure of a single super unit cell and the principle of eliminating the polarization sensitivity. The super unit cell consists of two columns of Al nanorods on the reflective panel, with a dielectric nanograting and a LC layer of thickness $d$ alternately arranged on top. The LC, a typical birefringent material, exhibits polarization-sensitive refractive index responses, namely the extraordinary and ordinary refractive indices ($n_e$ and $n_o$). In the figure, the LC molecules rotate within the $yz$-plane under a certain voltage, with a tilt angle $\theta_{LC}$ relative to the $y$-axis. To achieve polarization-insensitive modulation in LC, a combination of polarization conversion of the underlying metasurface and polarization-sensitive phase modulation of the LC is implemented. The resonant phase of asymmetric Al nanorods is utilized to form half-waveplates, with each nanorod's rotation angle set to either 45° or −45° along the $x$ axis. Unpolarized light can be regarded as a random combination of various polarized light types, such as linear, circular, and elliptical polarization, which can be described by $x$- and $y$-polarized bases. When $x$-polarized light is incident on the LC-covered columns, an initial propagation phase, denoted as $\varphi_{p1} = n_o \mathbf{k} d$ is established. $\mathbf{k}$ is the wave vector of the incident light. Subsequently, the metasurface-based half-wave plate

converts the polarization to $y$-polarization, while imprinting a phase $\varphi_m$ through metasurface resonance. As the light reflects and passes through the LC again, another propagation phase $\varphi_{p2} = n_{eff} \mathbf{k} d$ is established. The effective refractive index of the LC ($n_{eff}$) is changed from $n_e$ to $n_o$ and can be expressed as a function of the tilt angle $\theta_{LC}$ of the LC director from 0 to 90° controlled by voltage as $n_{eff} = \left( \frac{\cos^2 \theta_{LC}}{n_e^2} + \frac{\sin^2 \theta_{LC}}{n_o^2} \right)^{-1/2}$. Therefore, the overall phase modulation of the reflected light can be written as $\varphi = \varphi_m + \varphi_{p1} + \varphi_{p2}$. When $y$-polarized light is incident, the phase modulations during incidence and reflection are interchanged compared to $x$-polarized light, but the total phase modulation remains $\varphi = \varphi_m + \varphi_{p1} + \varphi_{p2}$, as shown in Fig. 2a. Consequently, for any incident polarized light, the dynamic phase modulations applied to its $x$ and $y$ components are identical—achieving polarization-insensitive phase modulation in the LC. For nanograting-covered columns, polarization-insensitive phase modulation can be naturally achieved with isotropic materials.

This polarization-insensitive property of the LC-covered columns can also be mathematically derived from the transmission matrix, which can be expressed as

$$
T = \begin{pmatrix} e^{i\varphi_{p1}} & 0 \\ 0 & e^{i\varphi_{p2}} \end{pmatrix} R(45°) \begin{pmatrix} e^{i\varphi_m} & 0 \\ 0 & e^{i(\varphi_m + \pi)} \end{pmatrix} R(-45°) \begin{pmatrix} e^{i\varphi_{p1}} & 0 \\ 0 & e^{i\varphi_{p2}} \end{pmatrix}
$$
$$
= e^{i(\varphi_m + \varphi_{p1} + \varphi_{p2})} \begin{pmatrix} 0 & 1 \\ 1 & 0 \end{pmatrix}
$$

$$(1)$$

where $R(\theta) = \begin{pmatrix} \cos\theta & -\sin\theta \\ \sin\theta & \cos\theta \end{pmatrix}$ is the rotation matrix. The transmission matrix is the product of the modulated phase term and the Pauli matrix. This indicates that arbitrary polarized incident light will be converted into a polarization that is orthogonal to the initial polarization after being modulated and reflected by the device (see also Supplementary Information Section 1). This conversion is accompanied by polarization-insensitive phase modulation of equal magnitude, which is consistent with the analysis in Fig. 2a.

Eliminating polarization sensitivity means that traditional polarization-selective methods for amplitude modulation (i.e., optical switching) are no longer applicable. To address this, we designed a dynamic subwavelength grating structure, formed by the upper nanograting and LC, to achieve high-contrast amplitude modulation. The contrast is defined as the ratio of the intensity of the "on" state to the intensity of the "off" state of optical switching. Figure 2b illustrates the specific structure design of the super unit cell, along with the corresponding phase profiles and resulting display intensities during the optical switching. A phase difference, $\triangle\varphi_x$, exists between the two columns of the super unit cell, determined by the underlying nanorods, and the nanograting and LC layers. This phase difference can be modulated to control the transition between propagating and evanescent waves when the nanograting period is subwavelength (see also Supplementary Fig. 1 and Supplementary Fig. 2). When $\Delta\varphi_x = (2j − 1)\pi (j = 1, 2, 3 \cdots)$, the reflected modulated light is completely suppressed due to destructive interference between neighboring columns. Conversely, as $\Delta\varphi_x$ deviates from $\pi$, the reflected light reappears, fully switching "on" when $\Delta\varphi_x = 2j\pi$. The geometric phase of the nanorods is used to create an initial phase difference, $\Delta\varphi_g = 2\Delta\theta = \pi$, where $\Delta\theta$ is the angle difference between the Al nanorods, meaning that the two columns of nanorods are perpendicular to each other. This precise geometric phase difference, introduced by the nanorod rotation, stabilizes the "off" state, enabling high-contrast performance. Additionally, the phase difference between the LC and the nanograting during incidence and reflection under

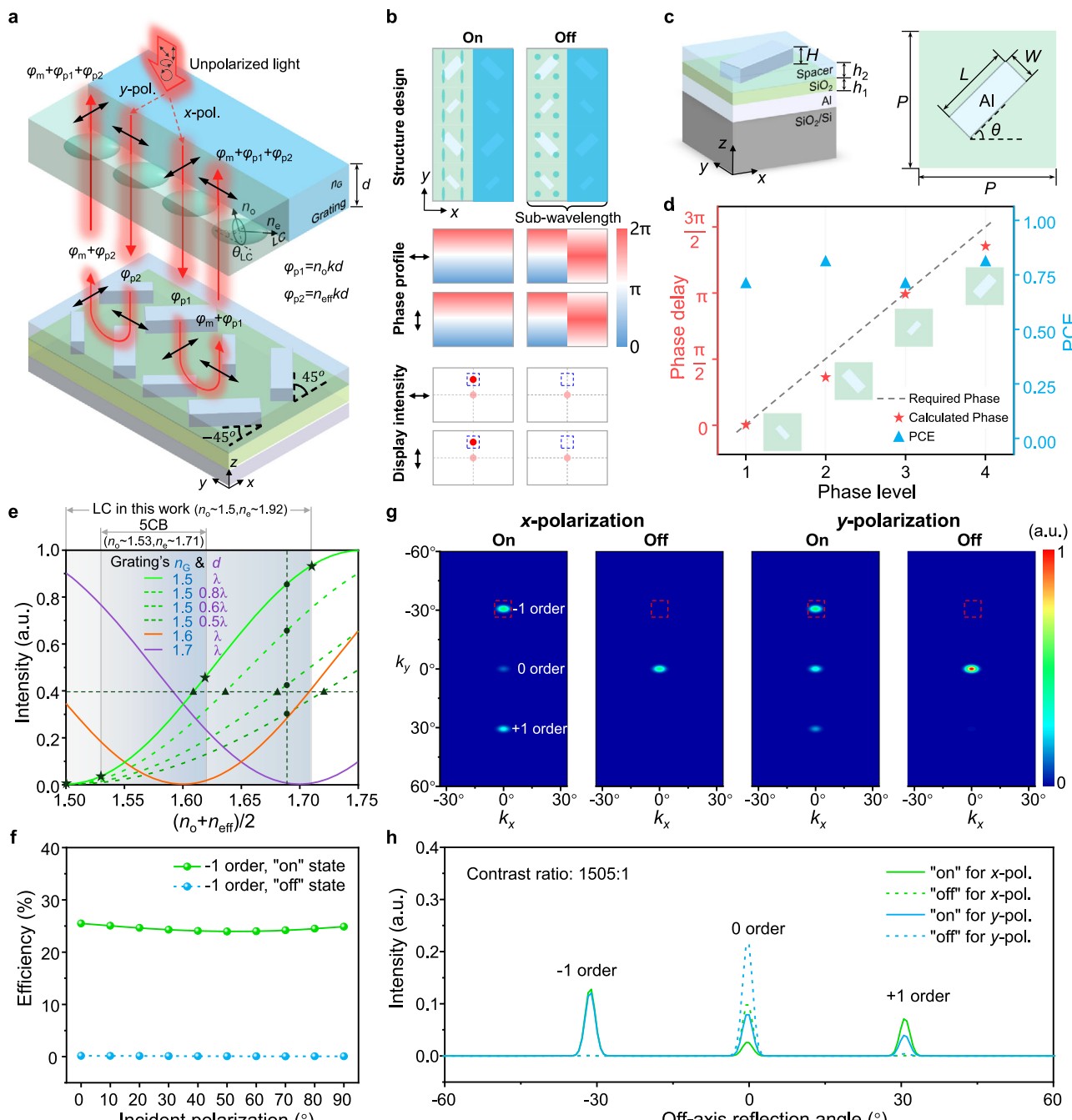

**Fig. 2 | Design and Simulation of the proposed polarization-insensitive meta-LCoS. a** The detailed structure of a single super unit cell and the principle of eliminating the polarization sensitivity of the polarization-insensitive meta-LCoS. **b** Schematic of the designed super unit cell, along with the corresponding phase profiles and the resulting display intensities. **c** The three-dimensional view and top view of the metasurface unit cell of the designed Al-SiO$_2$-Al MIM structure. The unit cells form a square lattice with a period of $P = 260$ nm at the operational wavelength of 532 nm. The thickness of each layer at the three operational wavelengths was optimized to the same parameters (200 nm for the Al mirror, $h_1 = 50$ nm for the SiO$_2$, $h_2 = 70$ nm for the spacer layer and $H = 50$ nm for the nanorods) for the purpose of facilitating fabrication. **d** Phase delay and polarization conversion efficiency for the different phase levels. Four phase levels are used in the design. The stars indicate the nanorods selected to construct the metasurface at the operational wavelength of 532 nm. **e** Calculated intensity of the off-axis reflection in response to changes in the refractive index and height of the LC and nanograting. **f** Simulated light efficiency of the off-axis reflection light at different incident polarization angles from the super unit cell with an operational wavelength of 532 nm. The light efficiency is calculated as the ratio of the optical intensity contained within three times the full width at half maximum of the diffraction order to the total optical intensity of the incident light. **g, h** Far-field simulation results (**g**) and the normalized intensity simulation curve (**h**) from polarization-insensitive meta-LCoS at the operational wavelength of 532 nm with an off-axis reflection angle of 30.8° in the y-direction.

arbitrary polarized or unpolarized light, $\Delta\varphi_p = \Delta\varphi_{p1} + \Delta\varphi_{p2} = (n_o + n_{eff} - 2n_G)\mathbf{k}d$, provides a dynamic modulation degree of freedom. By dynamically adjusting $\Delta\varphi_p$, the total phase difference $\Delta\varphi_x = \Delta\varphi_g + \Delta\varphi_p$ can be tuned from 0 to π, thus enabling the dynamic optical switching functionality.

To ensure high-contrast performance by separating the modulated light from the zero-order light, a linear phase gradient ($\frac{d\phi}{dy} = \frac{2\pi}{4P}$, where $P$ is the lattice period of single unit cell which are optimized to be less than half of the wavelength of the incident light) is also introduced along the y-direction to form an off-axis reflection point as the

switching point (see also Supplementary Fig. 3). The phase gradient is achieved by adjusting the geometry and rotation angle of the nanorod arrays, combining geometric and resonant phases. The angle of off-axis reflection light can be determined using the generalized Snell's law[27]. By combining the aforementioned design, we can theoretically achieve a polarization-insensitive, high-contrast, electrically driven optical switching with an off-axis reflection point, as illustrated in the two states shown in Fig. 2b.

### Numerical simulation and optimization

Figure 2c illustrates the composition of the underlying metasurface unit cell, which is designed as a metal-insulator-metal (MIM) structure of Al-SiO$_2$-Al. To prevent the infiltration of LC into the nanorods, the nanorods are coated with a spacer. Based on the proposed design methodology, the individual metasurface unit cell is simulated numerically to determine the optimal dimensions at three operational wavelengths of 465 nm, 532 nm and 633 nm (see Supplementary Information Section 4 for the detailed simulation setup). Phase control ($|\varphi_{me} - \varphi_{mo}| = \pi$) achieved through customized nanorod size leads to the highest polarization conversion efficiency (PCE). $\varphi_{me}$ and $\varphi_{mo}$ are the phase delays in the slow and fast axes of the nanorod, respectively. Considering the phase shift, PCE, and reflectivity of the metasurface unit cell, two types of nanorods with dimensions of 110 nm × 55 nm × 50 nm and 260 nm × 70 nm × 50 nm have been selected and arranged to form a half-wave plate with four phase levels at the operational wavelength of 532 nm, which then constitute the super unit cell, as illustrated in Fig. 2d. Each level incorporates the phase delay of 0, π/2, π, 3π/2, successively. The selected nanorods closely match the required phase profile and exhibit an average PCE of ~75%. Simulation results at operational wavelengths of 465 nm and 633 nm are shown in the Supplementary Fig. 4. The spectral response of the selected nanorods is also simulated, as shown in Supplementary Fig. 5.

For the design of the upper layer comprising LC and nanograting, the phase difference between the LC and nanograting, $\Delta\varphi_p = (n_o + n_{eff} - 2n_G)\mathbf{k}d$, should be controlled to achieve dynamic optical switching. Figure 2e illustrates the variation in the intensity of the off-axis reflection in response to changes in the refractive index and height of the LC and nanograting. The horizontal axis $(n_o + n_{eff})/2$ reflects the refractive index property of the LC under the applied voltage control. The thickness $d$ of the nanograting is set as $\lambda$, $0.8\lambda$, $0.6\lambda$, and $0.5\lambda$, accordingly, while the refractive index of the nanograting $n_G$ is varied to different values of 1.5, 1.6, and 1.7, accordingly. $\lambda$ is the wavelength of the incident light. As illustrated in the figure, when $n_G = (n_o + n_{eff})/2$, $\Delta\varphi_p = 0$, the off-axis reflection is completely extinguished due to the phase difference of nanorods, resulting in an "off" state critical for high-contrast displays. This indicates that the refractive index of the nanograting must be selected within the range between $n_o$ and $(n_o + n_e)/2$ to ensure the existence of the "off" state. Furthermore, with a constant refractive index for both the nanograting and LC, super unit cells composed of nanograting of varying heights can generate off-axis reflections of differing intensities (gray vertical dashed lines), without affecting the "off" state. This suggests that nanograting height minimally impacts contrast but influences the "on" state intensity, enhancing fabrication robustness. Additionally, for LC with higher birefringence, equivalent reflection intensities can be achieved with smaller nanograting heights (gray horizontal dashed lines), simplifying the fabrication process. Particularly, if $n_o$ is exactly equal to $n_G$, the "off" state is achieved when the maximum voltage is applied to the LC, significantly relaxing the voltage control requirements. Therefore, in this work, we opted to use a high-birefringence LC material ($n_e$ ~ 1.92 and $n_o$ ~ 1.5 at 25 °C for 532 nm) instead of the commonly used LC (e.g., 5CB) and chose PMMA ($n_G$ ~ 1.5) as the dielectric nanograting (see Supplementary Tbl. 1 for the parameters of high-birefringence LC). This design enables the device to be in the "on" state when no voltage is applied ($\Delta\varphi_p = 0.42\mathbf{k}d$) and an "off" state

when voltage is applied to cause $n_{eff} = n_o = n_G = 1.5$, resulting in $\Delta\varphi_p = 0$.

Figure 2f depicts the simulated light efficiency of the off-axis reflection point at different incident polarization angles from super unit cell with an operational wavelength of 532 nm. The efficiency of the off-axis reflection point remains substantially invariant under different polarization states of incident light, both in the "on" and "off" states of the device. This consistent performance further substantiates the polarization-insensitive characteristic of the design scheme. Figure 2g presents the far-field simulation results of an off-axis reflection point generated at the operational wavelength of 532 nm with an off-axis reflection angle of 30.8° in the $y$-direction, and the normalized intensity simulation curve is shown Fig. 2h. The off-axis angle is determined by the lattice period of the selected unit cell, and the red dashed box indicates the designed switching point. The simulation results demonstrate that the device is capable of producing a switching effect in both $x$- and $y$-polarization, with a high contrast ratio reaching 1505:1.

### Device fabrication and characterization

To prove the concept, we fabricated three devices operating at wavelengths of 465 nm, 532 nm, and 633 nm, respectively. The devices were mainly fabricated by electron beam lithography (EBL) overlay process, thermal evaporation, lift-off and LC packing, as illustrated in Fig. 3a (see Methods for details of the process)[28,29]. To simplify the fabrication process, the nanograting height was set at 300 nm. The long axes of LC molecules are aligned along the nanograting trenches due to the anchoring effect of the alignment layer on the glass cover and the nanograting. In the experiment, an LC layer with a thickness of 3 μm, which is much higher than the nanograting height, was established in order to reduce the difficulty of packaging and obtain a more homogeneous LC layer. Figure 3b, c shows the scanning electron microscopy (SEM) images of fabricated Al nanorod arrays and the subsequent PMMA nanograting fabricated using the EBL overlay process before LC infiltration, respectively. The effect of fabrication errors on device performance has also been investigated through simulation (see also Supplementary Figs. 6-8).

The optical measurement setup for characterizing the intensity and polarization insensitivity of the off-axis reflected light after LC packaging is illustrated in Fig. 3d. To ensure that the intensity of light with different linear polarization incident on the device is uniform, a polarizer and a quarter-wave plate are utilized to convert the light source into circularly polarized light. Figure 3e–g illustrates the intensity of the off-axis reflected light with the applied voltage under three operating wavelengths, respectively. As the applied voltage increases, the intensity of the off-axis reflected light decreases continuously (see also Supplementary Movies 1–3). The one-to-one correspondence between voltage and intensity allows for grayscale modulation in displays. This phenomenon can be attributed to the gradual change in the refractive index of LC from $n_e$ to $n_o$ under voltage. The experimental observations align with the calculated result depicted in Fig. 2e. The drive voltage is primarily influenced by the intrinsic characteristics of the LC and its thickness, and can be further optimized to suit the CMOS backplane. Intensity modulations with contrast ratios of 58.6:1, 81.3:1, and 62.4:1 were achieved at 465 nm, 532 nm, and 633 nm wavelengths, respectively. The contrast ratio is lower than that predicted by the simulation results due to the refractive index of PMMA nanograting does not precisely match the ordinary refractive index of LC. The fabricated PMMA nanograting exhibits a reduced effective refractive index due to porosity-induced structural variations (see also Supplementary Fig. 9). It is also affected by the near-field coupling of neighboring nanorods and the inaccuracies of the inherent fabrication process.

Figure 3h–j shows the intensity of off-axis reflected light at different incident polarization angles for devices operating at

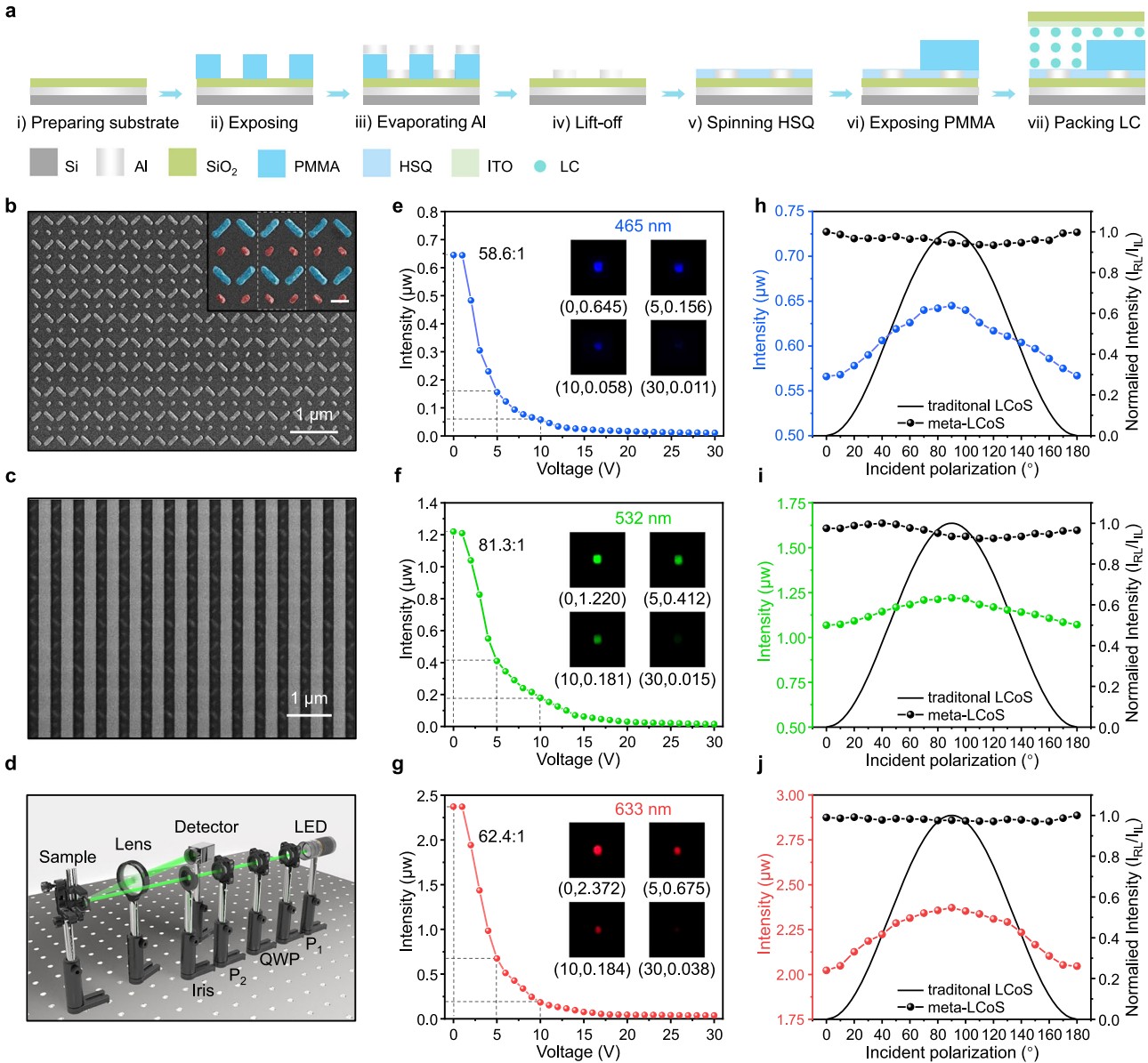

**Fig. 3 | Fabrication and characterization of the proposed polarization-insensitive meta-LCoS devices. a** Fabrication process of the polarization-insensitive meta-LCoS devices. **b, c** SEM images of fabricated Al nanorod arrays and devices prior to LC infiltration, respectively. Scale bar is 1 μm. Scale bar of the illustration is 200 nm. **d** Optical measurement setup for characterizing the off-axis reflected light. $P_1$ and $P_2$ represent polarizers, while QWP is a quarter-wave plate. Original 3D models adapted with permission from JCOPTIX.com. **e–g** Intensity of the off-axis reflected light as a function of the applied voltage at operational wavelengths of 465 nm, 532 nm and 633 nm, respectively. **h–j** Intensity of off-axis reflected light at different incident polarization angles for devices operating at wavelengths of 465 nm, 532 nm and 633 nm, respectively.

wavelengths of 465 nm, 532 nm and 633 nm, respectively. In light of the inherent fluctuations in intensity of incident light with different polarizations, the intensity ratio of the off-axis reflection light to the incident light ($I_{RL}/I_{IL}$, defined as the modulation efficiency) was characterized and normalized to highlight the polarization-dependent efficiency variation. The black curve represents the calculated modulation efficiency of conventional LCoS as a function of the incident polarization angle. For conventional LCoS devices, the efficiency decreases progressively as the polarization state deviates from the optimal alignment, ultimately reaching near-zero efficiency when the incident polarization becomes perpendicular to the LC orientation. In comparison to the conventional LCoS device, the fabricated meta-LCoS demonstrated a reduced intensity fluctuation, and the modulation efficiency remained essentially constant. This experimental result verifies the polarization-insensitive property of the device. The

switching time of this device, utilizing a 3 μm thick LC layer, is 40 ms and 65 ms for the "off" and "on" states, respectively, comparable to that of conventional LCoS device (see also Supplementary Fig. 10). It is noteworthy that only the LC molecules present in the PMMA trenches contribute to the amplitude modulation, thereby promoting a reduction in the LC thickness. Consequently, decreasing the LC thickness through advanced packaging technologies can significantly enhance switching speeds and reduce drive voltages. These improvements collectively lead to lower energy consumption in LCoS chips and support higher resolution displays.

## 64-pixel polarization-insensitive monochrome meta-LCoS dynamic display
Our concept can be further extended to displays with multi-pixel addressing control. Specifically, we fabricated three 64-pixel

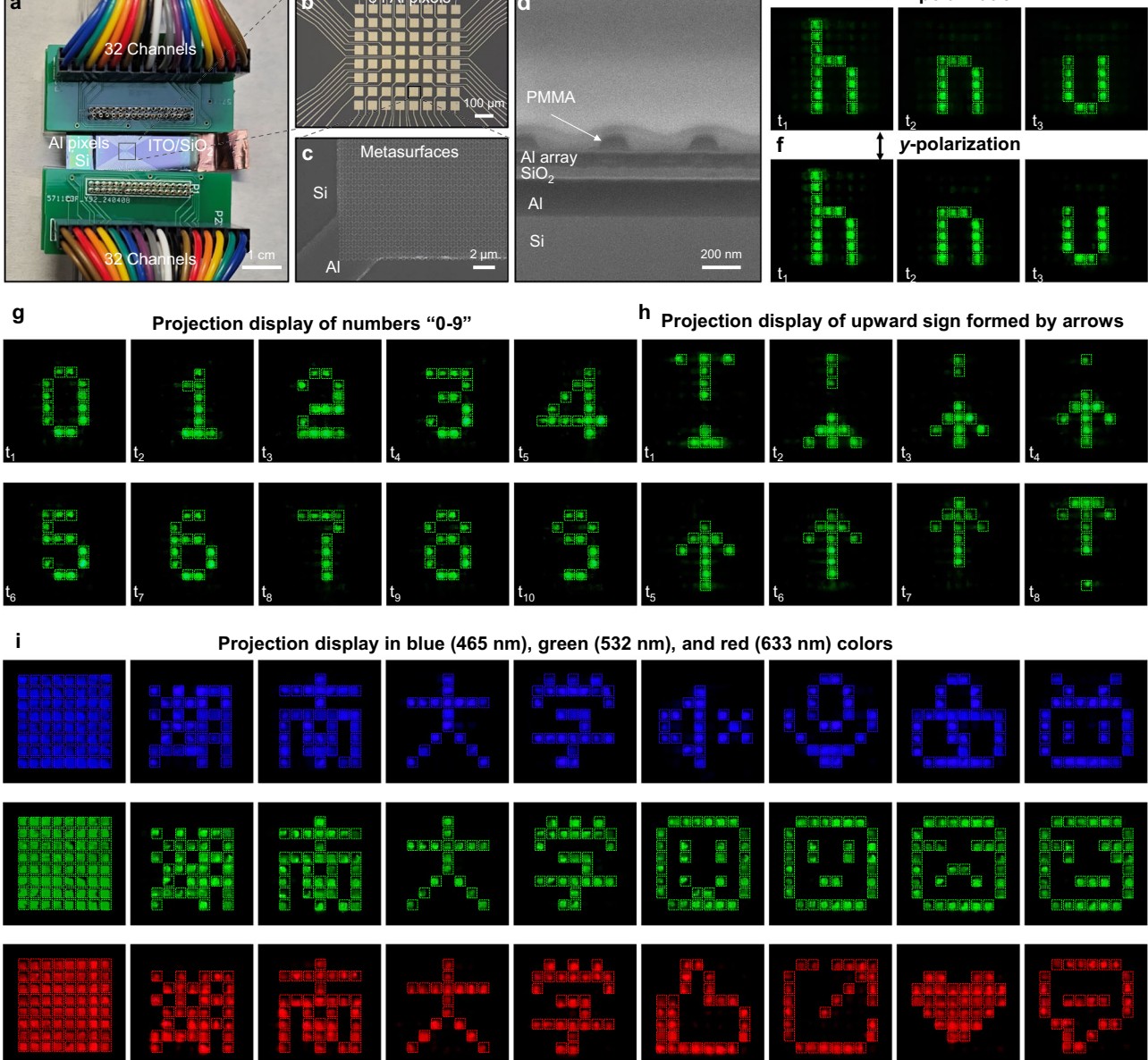

**Fig. 4 | Demonstration of 64-pixel polarization-insensitive monochrome meta-LCoS dynamic projection display. a** Photograph of the 64-pixel polarization-insensitive meta-LCoS device. The Al electrode leads are bonded to the PCB board contacts. Scale bar is 1 cm. **b** Optical microscope photograph of 64 Al pixels. Scale bar is 100 μm. **c** SEM image of the metasurface fabricated onto the Al pixels using the EBL overlay technique. Scale bar is 2 μm. **d** SEM image of the cross-sectional view of the metasurface following nanograting fabrication. Scale bar is 200 nm. **e**, **f** Dynamic projection display of the three letters "h", "n" and "u" in *x*-polarization and *y*-polarization, respectively. **g** Dynamic projection display of numbers "0–9". **h** Dynamic projection display of arrows. The image forms a dynamic upward sign. **i** Dynamic projection displays at operational wavelengths of 465 nm (blue), 532 nm (green) and 633 nm (red), respectively.

monochrome 2D addressable polarization-insensitive meta-LCoS prototype devices operating at wavelengths of 465 nm, 532 nm, and 633 nm, respectively, for projection display applications. Figure 4a shows a photograph of the final test device bonded to the leads of a printed circuit board (PCB). We first fabricated 64 square Al pixels, each measuring 50 μm, with a pixel pitch of 72 μm, serving as reflective mirrors, as depicted in the optical microscope photograph in Fig. 4b. The Al pixels were connected by leads and bonded to PCB contacts. Both the Al pixels and their leads were simultaneously exposed using a direct laser writing device. Subsequently, the metasurfaces were fabricated onto the pixels using the process outlined in Fig. 3a, as shown in the SEM image in Fig. 4c (see also Supplementary Fig. 11 and Fig. 12). Notably, the concept can be extended to practical applications with

smaller pixel sizes and higher resolutions through further optimization using the advanced CMOS process, as the generation of off-axis reflected light is solely related to the phase gradient period of the metasurface super unit cell. Figure 4d presents the SEM image of the cross-sectional view of the metasurface on the Al pixel electrode following the fabrication of the nanograting. Distinct layers can be distinguished in the cross-sectional view. Despite the deposition of a protective layer, the PMMA nanograting still shrinks because of high-energy ion-beam irradiation. Figure 4e, f shows the dynamic projection of the letters "h", "n" and "u" under *x*-polarized and *y*-polarized incident light, respectively (see also Supplementary Movie 4). The polarization-insensitive property of the device is further evidenced by the display effects observed under both *x*-polarization

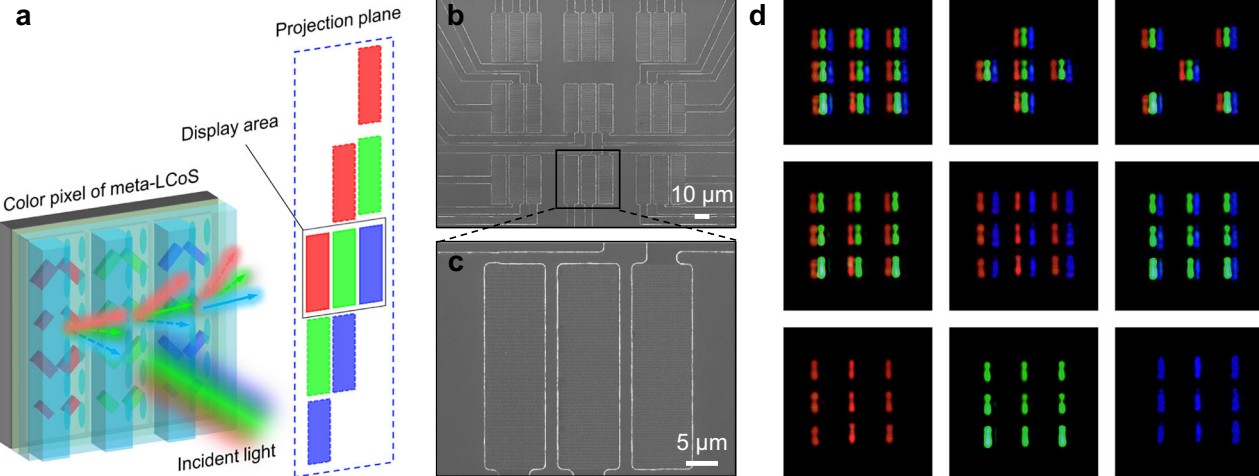

**Fig. 5 | Demonstration of polarization-insensitive monolithic color meta-LCoS dynamic projection display. a** Schematic of the proposed polarization-insensitive monolithic color meta-LCoS projection display. **b** Overall SEM image of the color pixels in the meta-LCoS device prototype. Scale bar is 10 μm. **c** Top view SEM images of the portion of the device marked with the black frame in **b**. Scale bar is 5 μm. **d** Projection display of color pixels. Each color pixel comprises red, green, and blue subpixels, each composed of nanorod arrays with specifically designed periods. These subpixels can be individually controlled by electrodes to produce images in various colors.

and *y*-polarization. To expand the capability of the concept, we conducted additional experiments using an LED source without polarizing elements, involving the dynamic projection display of numbers "0–9" and arrows, as shown in Fig. 4g, h (see also Supplementary Fig. 13 for testing setup and Supplementary Movie 5 and 6). Figure 4i shows the projection display of the devices operating at three wavelengths, 465 nm (blue), 532 nm (green) and 633 nm (red), respectively. The first image in each row depicts the projection onto a screen with all 64 pixels in the "on" state, as captured by the camera. The heterogeneity of the projection images may be attributed to a number of factors, including fabrication defects in the nanopillars, incomplete infiltration of the LC and differences in the driving voltage. To further exemplify the capabilities of our devices, Chinese characters "Hu", "Nan", "Da", and "Xue", representing Hunan University and other motifs, were showcased. (see also Supplementary Movie 7–9). For color projection display, we consider a three-chip architecture optical engine prototype employing beam-splitting dichroic prisms, which is based on the designed polarization-insensitive monochrome meta-LCoS. (see also Supplementary Fig. 14).

### 9-pixel polarization-insensitive monolithic color meta-LCoS dynamic display

To enhance the capabilities of the polarization-insensitive meta-LCoS, we demonstrate a meta-LCoS prototype that achieves full-color projection on a single chip through wavelength-specific off-axis angle engineering. The schematic of the device is illustrated in Fig. 5a. Each color pixel consists of three subpixels corresponding to red, green, and blue, constructed with Al nanorod arrays. These subpixels are individually controllable via Al electrodes, enabling the display of distinct color images. The periods of nanorods on the subpixels are precisely engineered to ensure a uniform off-axis projection angle for all three colors. Specifically, the red, green, and blue subpixels achieve a consistent 30.4° off-axis projection angle with nanorod array periods of 314 nm, 264 nm, and 230 nm, respectively. Owing to the dispersive properties, crosstalk light with varying off-axis angles may occur on the projection plane, necessitating spatial filtering using an iris to mitigate this effect (other monolithic color display scheme see also Supplementary Fig. 15). Figure 5b presents a SEM image of the color meta-LCoS device prototype, featuring nine color pixels, each with a subpixel size of 34 × 10 μm and a 2 μm gap between adjacent subpixels. Figure 5c shows a SEM image of one of the color pixels marked with the

black frame in Fig. 5b. Experimental results of the projected display, as depicted in Fig. 5d, confirm the efficacy of the high-contrast monolithic color display (see also Supplementary Movie 10).

## Discussion

In summary, we have demonstrated a meta-LCoS prototype chip that achieves polarization-insensitive, high-contrast monolithic full-color amplitude modulation. Its polarization-insensitive property and ability to achieve full-color display on a single chip will significantly simplify the configuration of the LCoS optical engine. The proposed device can also have a significantly reduced LC thickness due to the dynamic amplitude modulation by using only the LC molecules in the nanograting trenches. When the thickness of LC is sufficiently reduced, the grating is capable of producing uniform LC alignment without the need for any alignment layer, further simplifying device fabrication and reducing response time and drive voltage[30,31]. This design facilitates small pixel sizes and the suppression of inter-pixel crosstalk.

In addition, the concept can incorporate dielectric nanopillars to enhance efficiency and contrast further, while avoiding thermal generation in the structure. (see also Supplementary Fig. 16, Tbl. 2 and Fig. 17). Moreover, other active materials with refractive index variations can also be used in the proposed design methodology for various applications[32–34]. The performance of the devices will be further enhanced by nonlocal design with consideration of near-field coupling between nanorods[35–37] and advanced inverse design methods[38–42]. In comparison with DMD and conventional LCoS, the proposed device exhibits notable potential application advantages in terms of system simplicity, resolution and minimum pixel size (see also Supplementary Tbl. 3). Our devices will contribute to the development of next-generation customized high-resolution, high-contrast, low-cost projection and near-eye displays, and will also significantly advance the field of dynamic optical metasurfaces.

## Methods

### Numerical simulation

The responses of the device were designed and simulated using the finite-difference time-domain (FDTD, Ansys Lumerical FDTD) method. Critical cell parameters, including the length ($L$), width ($W$), thickness ($H$) and period ($P$) of the nanorods as well as the thickness of the SiO$_2$ ($h_1$) and spacer layer ($h_2$), need to be optimized to achieve proper phase, higher PCE and reflectivity. The PCE is defined as the ratio of the

optical power of the reflected light with orthogonal polarization to the incident optical power. See Supplementary Information Section 4 for detailed simulation settings.

## Device fabrication

The sample of single-pixel metasurfaces was fabricated as follows. A 5 nm chromium (Cr) adhesion layer and a 200 nm Al film were sputtered onto a 4-inch silicon wafer using ion-beam sputtering. Next, a 50 nm silica spacer layer was sputtered. The wafer was then diced into 15 × 15 mm substrates. The samples were mainly fabricated using the EBL overlay process. First, a structural layer composed of the metasurfaces and alignment marks was defined on the substrate using EBL. A 50 nm Al layer was deposited using thermal evaporation, followed by lift-off. Next, a 70 nm layer of hydrogen silsesquioxane (HSQ) was spin-coated onto the substrate and baked at 250 °C for 1 hour to remove the solvent. Subsequently, a 300 nm layer of PMMA resist layer was coated onto the substrate. The PMMA nanograting was fabricated by EBL and aligned with the metasurfaces of the first layer using the Al markers. The preparation of the multi-pixel samples necessitated the fabrication of Al electrode pixels as a preliminary step. Electrode patterns and alignment marks were defined on the substrate using laser direct writing (LDW) lithography on a 30 × 30 mm silicon substrate with 285 nm oxide, as shown in Fig. S6. The remaining steps are identical to those used for creating single-pixel samples, with the exception that the Al electrode pads must be carefully protected throughout the fabrication process.

## LC cell packaging

A polyimide layer was spun on an ITO-coated glass substrate and subsequently cured on a hot plate at 100 °C for 30 min. The LC orientation directions were obtained by unidirectional rubbing of the polymer layer. After that, the glass substrate was adhered to the metasurface samples using UV-curable adhesive (NOA 81) containing 3 μm glass spacer beads. Following the application of the UV-curable adhesive, the cell was heated to 70 °C and infiltrated with LC. Once the cell has cooled to room temperature, the edges are sealed with UV-curable adhesive.

## Data availability

The authors declare that the data supporting the findings of this study are available within the paper, its Supplementary Information Files and Source data. Source data are available at https://doi.org/10.6084/m9.figshare.30071761.

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

## Acknowledgements

The authors thank Dr. Fan Fan and Dr. Zenghui Peng for their guidance on the LC packaging process. We acknowledge the financial support from the National Natural Science Foundation of China (Grant no. 62275078, Y.H. and Grant no. 52425508, H.D.), National Key Research and Development Program of China (Grant No. 2023YFB2806504, Y.H.), the Science and Technology Innovation Program of Hunan Province (Grant no. 2023RC3101, Y.H.) and Shenzhen Science and Technology Program (Grant no. JCYJ20220530160405013, H.D.).

## Author contributions

X.O. and Y.H. contributed equally to this work. Y.H., X.O. and S.Lou proposed the idea. X.O., D.Y., S. Lou and J.L. conceived and carried out the design and simulation. X.O., D.Y., Z.S., W.W. and M.L. prepared the metasurface samples. X.O. and S.Liu carried out the LC packaging. X.O., S.Liu and Y.H. conceived and performed the measurements. X.O., Y.H., J.L., T.C., N.L. and H.D. analyzed the data and co-wrote the manuscript. Y.H. and H.D. supervised the overall project. All authors discussed the results and provided comments on the manuscript.

## Competing interests

The authors declare no competing interests.
