## [Transparent Peer Review file · Nature Communications]

Meta-Optics Redefines Microdisplay: Monolithic Color LCoS without Polarization Dependency

Corresponding Author: Dr Yueqiang Hu

Version 0:

Reviewer comments:

Reviewer #1

(Remarks to the Author)

This paper presents a novel advancement in liquid crystal on silicon (LCoS) microdisplay technology by integrating metasurfaces to create a polarization-insensitive, monolithic full-color LCoS device. Its polarization insensitivity and monolithic color capability are groundbreaking, though technical metrics like contrast and switching speed need enhancement to competing commercial standards. Addressing the below questions will strengthen the manuscript by clarifying mechanisms, and outlining paths for improvement, ensuring its impact on the field of dynamic optical metasurfaces and display technologies.

1. This paper states that polarization insensitivity is achieved through metasurface polarization conversion and LC phase modulation. Could you explain more on how the metasurface converts arbitrary polarization into a specific state, and how the LC modulates phase consistently across all polarization states? A detailed schematic or mathematical derivation would clarify this synergy.
2. The authors determined for a high-birefringence LC ($n_e \approx 1.92$, $n_o \approx 1.5$) over standard 5CB. What motivated this choice beyond refractive index matching with PMMA? How does it impact switching speed, voltage requirements, and contrast compared to 5CB?
3. Since LC thickness directly affects the phase difference, could you provide an example illustrating how changing the thickness from $3 \mu\text{m}$ to another value alters the phase difference and key performance metrics like contrast ratio or switching time? How did you ensure that the $3 \mu\text{m}$ LC layer remains uniform across the entire meta-LCoS device during fabrication? What are the concerns of any thickness non-uniformity on device performance?
4. Experimental contrast ratios are significantly lower than simulated values (e.g., 81.3:1 vs. 1505:1 at 532 nm). Can you quantify these effects and propose solutions (e.g., alternative nanograting materials, process refinements)?
5. The meta-LCoS devices were fabricated using electron beam lithography overlay. How much overlay errors are acceptable without affecting the device performance?
6. For practical applications like projectors or AR/VR displays that demand high resolutions (e.g., 1080p or 4K), what challenges arise in scaling the meta-LCoS technology from its current prototypes (64-pixel monochrome and 9-pixel color)? How are issues such as pixel crosstalk, fabrication yield, and maintaining contrast at smaller pixel sizes addressed? Does the metasurface layer introduce unique thermal management challenges, and if so, what strategies could mitigate them?
7. Switching times of 40 ms and 56 ms are reported. What factors (e.g., LC viscosity, layer thickness, anchoring strength) most influence these times, and what strategies (e.g., thinner LC layers, alternative materials) could reduce them to below 10 ms?
8. For the monolithic color meta-LCoS, how does the color gamut and accuracy compare to traditional LCoS systems? Please provide quantitative metrics (e.g., CIE coordinates) or discuss crosstalk improvement beyond spatial filtering.
9. Supplementary Section 8 explores dielectric nanopillars but decides for aluminum due to fabrication challenges. What specific improvements (e.g., material choice, lattice design) could make them viable?

Reviewer #2

(Remarks to the Author)

The authors demonstrate a monolithic color meta-LCoS prototype that integrates dual-layer metasurfaces to achieve

polarization-insensitive, full-color amplitude modulation. The pixelated monochrome and color prototype capable of dynamically projecting diverse patterns under unpolarized illumination can be achieved by forming dual-layer subwavelength gratings composed of metal and dielectric nanostructures on pixelated LCoS backplane. The work demonstrates significant advancements in simplifying optical engines for projection and AR/VR displays. I thus recommend that this manuscript be accepted for publication in Nature Communications. The following comments and issues, once properly addressed, should help improve the manuscript.

1. In contrast to DMD and conventional LCoS schemes, the authors mentioned that polarization-insensitive meta-LCoS can achieve simple projection displays system with unpolarized source, high contrast and high resolution. I agree with their motivations for studying polarization-insensitive meta-LCoS, but total efficiency and CMOS compatibility for integration of metasurface should also be carefully considered. More comments and discussions are suggested to strengthen the manuscript and make it more convincing and impactful.

2. For the design of meta-LCoS shown in Fig. 2, I wonder why the authors chose Al metasurface, SiO₂ and Al electrode to form the reflective HWPs and subwavelength gratings. The selection of 4 - level phase and ordinary PCR (nearly 75%) in Fig. 2d results in low total efficiency (nearly 10%) in the targeted diffractive order shown in Fig. 2h. Besides, please supply the clear definition of contrast ratio.

3. The design choice of a 30.8° off-axis reflection angle in the polarization-insensitive meta-LCoS (Fig. 2) requires clarification: (i) What specific projection system requirements necessitate this angular deviation from conventional on-axis configurations? (ii) How does this parameter affect the optical efficiency and contrast ratio within the targeted diffraction order? (iii) Could the implemented angular offset potentially compromise the device's performance metrics as shown in Fig. 2h?

4. For the fabrication process of the polarization-insensitive meta-LCoS shown in Fig. 3a, I wonder whether the "Spinning HSQ" process is reliable, because the baked temperature of HSQ is 250 °C. It would be better to address detection protocols for aluminum oxidation in both the electrode and metasurface components. Oxidation-induced defects are likely to degrade the device's electrical conductivity and optical performance, as evidenced by potential impacts on diffraction efficiency and contrast ratio metrics.

5. For manufacturing tolerances, I've noticed that nanorod dimensions (e.g., ±5 nm variations) and PMMA grating profiles are likely to influence device performance. Has a sensitivity analysis been performed to establish acceptable fabrication tolerances?

6. For the characterization of the polarization-insensitive meta-LCoS shown in Figs. 3e-3g, the authors explained that the lower contrast ratio compared to simulated results due to inherent fabrication process and refractive index mismatch between PMMA nanograting and ordinary refractive index of LC. Have the authors considered the oxidation of Al electrode and Al metasurface? Please also discuss mitigation strategies.

7. The drive voltage that a CMOS backplane can apply is generally within the range of 0 to 5V. For the experimental results shown in Figs. 3e-3g, the authors explained that the drive voltage is primarily influenced by the intrinsic characteristics of the LC and its thickness. The optimization of intrinsic material parameters (e.g., refractive index anisotropy, elastic constants) within high-birefringence liquid crystal (LC) materials requires systematic investigation to achieve target modulation depth through tailored electro-optic response engineering. More in-depth discussions are needed to better support the conclusions.

8. A systematic evaluation is needed to establish device performance benchmarks through a quantitative comparison of polarization-dependent total efficiency between conventional LCoS and meta-LCoS configurations, as shown in Figs. 3h-3j, across varying incident polarization states.

9. For the SEM images of the device shown in Figs. 4b-d, the active region is at the microscale, and the HSQ spacer layer appears to have been flattened before PMMA nanograting fabrication. Given the flatness and uniformity of the spacer layer, I question whether the proposed 'Spinning HSQ' scheme is suitable for LCoS chips in centimeter-scale applications beyond 2K resolution. Could the authors address potential fabrication challenges for high-resolution meta-LCoS devices?

10. Regarding the dynamic projection displays driven by the polarization-insensitive meta-LCoS shown in Fig. 4i, the projection images of the 64 pixels in the "on" state appear to exhibit heterogeneity. Can the authors provide an explanation for this observation and specifically comment on its implications for high-resolution meta-LCoS devices?

11. The 9-pixel color prototype, while a proof of concept, is insufficient to demonstrate high-resolution color projection. The scalability to higher pixel densities (e.g., 8K) remains unaddressed. How is spatial filtering optimized for multi-wavelength alignment?

12. For practical display applications, CMOS compatibility is crucial for meta-LCoS devices. I noticed that the authors have begun to discuss the design of dielectric nanopillars for meta-LCoS in the supplementary materials. Can the authors elaborate on potential challenges in device fabrication using the CMOS-compatible process to incorporate dielectric nanopillars?

Reviewer #3

(Remarks to the Author)

The work entitled "Meta-Optics Redefines Microdisplay: Monolithic Color LCoS without Polarization Dependency" by Ou et al, introduces a novel meta-LCoS microdisplay that integrates dual-layer metasurfaces onto a conventional LCoS panel. The approach utilizes a bottom aluminium metasurface to convert polarization, enabling liquid crystal (LC) to modulate both polarization states equally. To address the absence of polarization-dependent amplitude control, the authors incorporate an upper metasurface layer made of PMMA/LC nanogratings. This layer modifies interference of outgoing light in the lateral direction, allowing dynamic control of the phase difference between the nanograting and LC regions. This adjustment facilitates constructive and destructive interference, effectively managing the "On" and "Off" states of each pixel. By merging metasurface-enabled polarization conversion with voltage-controlled LC phase modulation, the device eliminates the inherent polarization sensitivity of traditional LCoS systems. This innovation tackles a critical limitation of conventional LCoS displays—their reliance on linearly polarized light and the need for bulky, costly polarizing optics. The study experimentally validates the concept with a 64-pixel monochrome prototype operating at three key wavelengths (465 nm, 532 nm, and 633 nm). Additionally, a three-chip architecture optical engine prototype is proposed for color projection displays. However, the 9-pixel single-chip full-color prototype exhibits significant crosstalk, attributed to the suboptimal resonant performance of the bottom metasurface. Despite this, the proposed concept and demonstration of a monolithic color display hold substantial significance. This work marks a notable advancement in microdisplay technology, introducing a monolithic, polarization-insensitive meta-LCoS with the potential to streamline optical systems and enhance performance in next-generation AR/VR, HUD, and projection display applications.

Therefore, I recommend publication pending revisions. Below are some points that could be addressed to enhance the manuscript's clarity:

1. While the integration of metasurface technology with LCoS is promising, the manuscript would benefit from a clearer comparison (such as a table) with existing approaches and a detailed discussion on how this work uniquely advances the state-of-the-art in terms of performance, device complexity, and integration capabilities.
2. Although the full-color prototype represents a significant step forward, the proposed single-chip color display is limited by its broad spectral response, requiring a spatial filter and leading to reduced light utilization efficiency. The current design requires a resonant metasurface, yet challenges remain with the metallic structure used. Further details on the resonance performance (such as spectral response) of the selected unit cells are necessary. Including simulation results and a discussion on the current limitations, particularly regarding crosstalk caused by metallic structures, would significantly strengthen the analysis.
3. The authors state that in the "Off" state, dark pixels result from light being coupled into the evanescent wave domain. In the characterized sample, could this lead to substantial heating within the device? Might it impact LC performance or the integrity of the LC encapsulation? The authors should provide commentary on these potential effects.
4. In simulation, green and red incidences show similar maximum reflected intensity (fig 2e). However, the experimental results in fig 3f and 3g show the maximum intensity for red being nearly double that of green. What accounts for this discrepancy?
5. The authors attribute contrast ratio difference (1505:1 in simulation and 81.3:1 in experiments) to a mismatch in the refractive index of the PMMA nanograting with the LC. Isn't the simulation also use PMMA/LC as the grating material? The same condition should be used for fair comparison. What are potential ways to improve this.

Version 1:

Reviewer comments:

Reviewer #1

(Remarks to the Author)

The authors have effectively addressed the proposed problem. I recommend this manuscript for publication in Nature Communications.

Reviewer #2

(Remarks to the Author)

The reviewer is satisfied with the revisions made to the manuscript and therefore recommends its acceptance for publication in Nature Communications.

Reviewer #3

(Remarks to the Author)

In the revised manuscript, the authors have addressed all of my concerns and comments with clarity and depth. The work is highly novel and highlights the advantages of integrated meta-optics for compact, robust, and multifunctional device applications. I am now pleased to recommend the manuscript for acceptance and publication.

Reviewer 1:

Comments:

This paper presents a novel advancement in liquid crystal on silicon (LCoS) microdisplay technology by integrating metasurfaces to create a polarization-insensitive, monolithic full-color LCoS device. Its polarization insensitivity and monolithic color capability are groundbreaking, though technical metrics like contrast and switching speed need enhancement to competing commercial standards. Addressing the below questions will strengthen the manuscript by clarifying mechanisms, and outlining paths for improvement, ensuring its impact on the field of dynamic optical metasurfaces and display technologies.

Response. We thank the reviewer for the evaluation of our work. We appreciate the detailed comments. We are happy to receive such excellent advice for improving the manuscript.

Comments 1. *This paper states that polarization insensitivity is achieved through metasurface polarization conversion and LC phase modulation. Could you explain more on how the metasurface converts arbitrary polarization into a specific state, and how the LC modulates phase consistently across all polarization states? A detailed schematic or mathematical derivation would clarify this synergy.*

Response. We thank the reviewer for the comments. The following section presents a detailed explanation of the polarization conversion mechanism of the metasurface and the phase modulation of the LC, as well as their synergy in achieving polarization insensitivity.

Fig. R1.1 The principle of achieving polarization-insensitive modulation.

The metasurface unit structure of this device and the principle of achieving polarization-insensitive modulation are illustrated in Fig. R1.1. Unpolarized light can be regarded as a random combination of various polarized light types, such as linear, circular, and elliptical polarization, which can be described using two mutually perpendicular linear polarization bases. Therefore, the figure analyzes x -polarized and y -polarized light separately to demonstrate the scenario of unpolarized light incidence.

First, the LC is aligned such that its fast axis is parallel to the x -axis. When a voltage is applied, the LC molecules rotate within the yz -plane. When x -polarized light is incident on the LC-covered column, an initial propagation phase, denoted as $\varphi_{p1} = n_o k d$ (where n_o is the refractive index along the LC's fast axis), is established. Subsequently, the

metasurface-based half-wave plate converts the polarization to y -polarization while imprinting a phase φ_m through metasurface resonance. As the light reflects and passes through the LC again (now as y -polarized light), another propagation phase $\varphi_{p2}=n_{\text{eff}}kd$ is generated, where n_{eff} is an effective refractive index dependent on the rotation angle of the LC molecules. Thus, the total phase modulation of the reflected light can be expressed as $\varphi=\varphi_m+\varphi_{p1}+\varphi_{p2}$.

When y -polarized light is incident, the phase modulations acquired during the incident and reflected passes through the LC are interchanged, but the total phase modulation remains $\varphi=\varphi_m+\varphi_{p1}+\varphi_{p2}$. Consequently, for any incident polarized light, the dynamic phase modulations applied to its x and y components are identical—achieving polarization-insensitive phase modulation. For the nanograting-covered columns, polarization-insensitive phase modulation is naturally realized using isotropic materials.

In summary, this method ingeniously leverages three essential elements: (1) the dual processes (incident and reflected) of light in a reflective configuration, (2) polarization conversion via the metasurface, and (3) the birefringence of LC. The synergy of these elements is indispensable for realizing polarization insensitivity.

This polarization-insensitive property of the LC-covered columns can be also mathematical derived from the transmission matrix which can be expressed:

$$T = \begin{pmatrix} e^{i\varphi_{p1}} & 0 \\ 0 & e^{i\varphi_{p2}} \end{pmatrix} R(45^\circ) \begin{pmatrix} e^{i\varphi_m} & 0 \\ 0 & e^{i(\varphi_m+\pi)} \end{pmatrix} R(-45^\circ) \begin{pmatrix} e^{i\varphi_{p1}} & 0 \\ 0 & e^{i\varphi_{p2}} \end{pmatrix} \\ = e^{i(\varphi_m+\varphi_{p1}+\varphi_{p2})} \begin{pmatrix} 0 & 1 \\ 1 & 0 \end{pmatrix}$$

where $R(\theta)=\begin{pmatrix} \cos \theta & -\sin \theta \\ \sin \theta & \cos \theta \end{pmatrix}$ is the rotation matrix. The electric field of the emitted light can be expressed as $E_o = TE_i$. E_i denotes the electric field of the incident light.

When linearly polarized light is incident:

$$E_i = \begin{pmatrix} \cos \alpha \\ \sin \alpha \end{pmatrix}, E_o = TE_i = e^{i(\varphi_m + \varphi_{p1} + \varphi_{p2})} \begin{pmatrix} \sin \alpha \\ \cos \alpha \end{pmatrix}$$

When circularly polarized light is incident:

$$E_i = \begin{pmatrix} 1 \\ \pm i \end{pmatrix}, E_o = TE_i = e^{i(\varphi_m + \varphi_{p1} + \varphi_{p2})} \begin{pmatrix} \pm i \\ 1 \end{pmatrix}$$

When elliptically polarized light is incident:

$$E_i = \begin{pmatrix} Ae^{i\varphi_x} \\ Be^{i\varphi_y} \end{pmatrix}, E_o = TE_i = e^{i(\varphi_m + \varphi_{p1} + \varphi_{p2})} \begin{pmatrix} Be^{i\varphi_y} \\ Ae^{i\varphi_x} \end{pmatrix}$$

The matrix of the emitted light shows that the incident light with different polarizations is converted into light with orthogonal polarization after modulation and reflection by the device, accompanied by the same phase modulation independent of polarization, which is consistent with the previous analysis results.

Fig. 2a and the description of the principle in the manuscript have been updated for improved clarity. Furthermore, additional mathematical derivations have been included in the supplementary materials to enhance the comprehensiveness of the analysis.

Comments 2. *The authors determined for a high-birefringence LC ($n_e \approx 1.92$, $n_o \approx 1.5$) over standard 5CB. What motivated this choice beyond refractive index matching with PMMA? How does it impact switching speed, voltage requirements, and contrast compared to 5CB?*

Response. We thank the reviewer for the comments.

Fig. R1.2 Calculation of LC and nanograting parameters on the intensity of off-axis reflection.

The choice of a high-birefringence LC (with $n_e \approx 1.92$ and $n_o \approx 1.5$) over standard 5CB LC for this device is primarily motivated by the following factors:

First, the n_o of high-birefringence LC is equal to the refractive index of PMMA nanogratings, allowing for a stable “off” state at a maximum voltage. Upon the incidence of arbitrarily polarized or unpolarized light, the dynamic phase difference $\Delta\varphi_p = \Delta\varphi_{p1} + \Delta\varphi_{p2} = (n_o + n_{\text{eff}} - 2n_G)kd$ arises from the upper LC layer and the nanograting during the incident and reflected passes. The n_{eff} of LC can vary from n_e to n_o as the rotation angle of the LC molecules changes. When $\Delta\varphi_p = 0$, the device achieves the “off” state through destructive interference caused by the geometric phase difference carried by the two lower metasurface columns. Therefore, to ensure the existence of the “off”

state (i.e., a solution for $\Delta\varphi_p = 0$), the refractive index of the nanograting (n_G) must be selected within the range from n_o to $(n_o + n_e)/2$. This is visualized more intuitively in Fig. R1.2, which illustrates the analysis of how the LC material properties and nanograting parameters affect the device performance. The figure clearly shows the achievable switching performance with different LC materials, as well as varying refractive indices and heights of the nanograting. However, if the refractive index of the nanograting material is chosen to be a value between n_o and $(n_o + n_e)/2$, for example, selecting 5CB as the LC with $n_G = 1.6$, the “off” state occurs at a specific LC orientation where the effective refractive index (n_{eff}) lies between n_e and n_o . This requires highly precise voltage control to stabilize the “off” state. Conversely, if n_o is exactly equal to n_G , the “off” state is achieved when the maximum voltage is applied to the LC, significantly relaxing the voltage control requirements. Therefore, the selected high-birefringence LC, with n_o matching the refractive index of the PMMA, is chosen to meet this requirement. Of course, for other LC materials, a nanograting with a refractive index matching their respective n_o values could also be selected as an alternative.

Second, a higher-birefringence LC enables a larger phase modulation for the same thickness, allowing an equivalent reflection intensity to be achieved with a smaller nanograting height (as indicated by the horizontal gray dashed line in Fig. R1.2). This also means that a thinner LC layer can be used to realize the device functionality—only the LC layer with the same height as the nanograting participates in the amplitude modulation of the device. This not only reduces the fabrication difficulty of the nanograting but also facilitates the wetting of the grating by the LC and weakens the

anchoring effect of the grating on the LC molecules.

Regarding the impact on switching speed, voltage requirements, and contrast, the high-birefringence LC material has several advantages. In terms of switching speed, since only the LC with heights matching that of the nanograting contribute to the device's amplitude modulation, a higher birefringence enables a reduced LC layer thickness while achieving the same required modulation phase. Consequently, this leads to a significant enhancement in the device's switching speed. In our experiments, we observed that the switching times for the "off" and "on" states were 40 ms and 56 ms, respectively. This is because a 3 μm thick LC, which is much higher than the grating height (300 nm), was used in the experiment to reduce the difficulty of encapsulation and achieve a more uniform LC layer. Regarding voltage requirements, as mentioned earlier, the high-birefringence LC can theoretically allow for a reduction in the required driving voltage due to the thinner layer needed to achieve the same phase modulation. Regarding contrast, contrast is primarily determined by the "off" state brightness, with an ideal "off" state condition requiring a precise phase difference of zero between the LC and grating. Compared to 5CB, the n_o of high-birefringence LC is more compatible with the refractive index of PMMA, which is conducive to achieving higher contrast.

We have revised Fig. 2e and refined the narrative section on LC selection in the manuscript to ensure a more rigorous and clear description.

Comments 3. *Since LC thickness directly affects the phase difference, could you provide an example illustrating how changing the thickness from 3 μm to another value alters the phase difference and key performance metrics like contrast ratio or switching time? How*

did you ensure that the 3 μm LC layer remains uniform across the entire meta-LCoS device during fabrication? What are the concerns of any thickness non-uniformity on device performance?

Response. We thank the reviewer for the comments.

1. Influence of LC thickness on phase difference and performance

As illustrated in Fig. R1.1, only LC situated at the same height as the nanograting contributes to the regulation of device switching. LC positioned above the nanograting does not influence the phase difference between the LC column and the nanograting column. The contrast depends on this phase difference, which remains unaffected by LC thickness as long as LC thickness $d \geq$ grating height (300 nm). Thus, contrast is maintained. Additionally, the response time $t \propto d^2$ (due to viscoelastic relaxation), so a 3× thickness reduction theoretically enhances speed by 9×. However, in the experiment, in order to reduce the difficulty of packaging and achieve a more uniform LC layer, a LC layer with a thickness of 3 μm, which is much higher than the nanograting height (300 nm), was established.

2. Uniformity control for 3 μm LC layer

To ensure the 3 μm LC layer remains uniform across the entire meta-LCoS device during fabrication, it is necessary to package the LC in an absolutely clean environment (even a vacuum environment for higher requirements) and apply a uniform frame glue containing spacers of the same size to the edges of the substrate. A frame adhesive with low shrinkage and low stress is selected, and uniform pressure is applied to the substrate during the curing process of the frame adhesive to ensure the uniformity of the LC filling

space. After the LC cell was applied, we conducted thorough inspections using appropriate metrology tools to verify the uniformity of the layer. Any deviations were addressed promptly to ensure the quality of the device. After obtaining the LC cell, the thickness of the cell is also measured to verify the uniformity of the LC layer. During the LC package process, the LC cell is heated to 70°C to increase the fluidity of the LC, ensuring that it can uniformly wet the nanograting.

3. Impacts of thickness non-uniformity

Thickness non-uniformity in the LC layer can result in several issues that affect device performance. Variations in thickness can cause inconsistent phase modulation across the device, leading to uneven intensity and grayscale distortion in display applications. Furthermore, thickness non-uniformity may induce variations in switching times across the device, leading to temporal inconsistencies during dynamic operation. Thickness variations can also lead to differences in the required driving voltage across the device, complicating the electrical control. In our study, we carefully controlled the fabrication process to minimize thickness non-uniformity and ensure the consistent performance of the meta-LCoS device.

We have refined the discussion of the effect of LC thickness on device performance in the manuscript to ensure the completeness and rigor of the paper.

Comments 4. *Experimental contrast ratios are significantly lower than simulated values (e.g., 81.3:1 vs. 1505:1 at 532 nm). Can you quantify these effects and propose solutions (e.g., alternative nanograting materials, process refinements)?*

Response. We thank the reviewer for the comments.

The experimentally measured contrast ratios at operational wavelengths exhibit deviations from simulated predictions, primarily attributable to refractive index discrepancies of PMMA nanograting and fabrication-induced structural imperfections.

The first reason for this discrepancy is that the refractive index of the manufactured PMMA nanograting differ from the design values. In contrast, the refractive index of the two materials is set to be equal in the simulation. This mismatch in the experimental device results in incomplete destructive interference in the “off” state. As illustrated in Fig. R1.3a (i.e. Fig.4 in the reference, DOI: 10.1364/AO.54.00F139), the refractive index of PMMA ranges between 1.49 and 1.51 across the visible region. However, the fabricated PMMA nanograting exhibits a reduced effective refractive index (approximately 1.48 at $\lambda = 532$ nm, as quantified through spectroscopic ellipsometry in Fig. R1.3b) due to porosity-induced structural variations.

Fig. R1.3 The refractive index of PMMA used in the simulation (a) and in the experiment (b).

To quantitatively assess the impact of refractive index variations on device contrast, we have performed numerical simulations with a nanograting refractive index of $n_G = 1.48$. As shown in Fig. R1.4, this configuration yields a simulated contrast ratio of 136:1, which is slightly higher than the contrast ratio achieved in the experiment.

Fig. R1.4. Numerical contrast ratio with the PMMA refractive index $n_G = 1.48$ at the operating wavelength of 532 nm.

Furthermore, fabrication and packaging imperfections contribute significantly to contrast ratio degradation. Structural deviations—including dimensional inaccuracies, morphological defects, and sidewall steepness variations—compromise the designed phase gradient fidelity, thereby attenuating destructive interference in the “off” state. During encapsulation, incomplete LC infiltration induces microbubble formation, further distorting the phase gradient of the device. These combined effects reduce the maximum achievable contrast compared to ideal simulated conditions.

To mitigate these effects and improve the contrast ratio, we propose the following solutions. To address the refractive index mismatch between PMMA and the n_o of the LC, strategic doping of high-index nanoparticles (e.g., TiO_2 , $n \approx 2.5$ at 532 nm) into the PMMA presents a viable solution (reference DOI: 10.1002/app.32567, 10.1016/j.ijleo.2014.04.077). The use of another nanograting materials with refractive index that more closely match the n_o of the LC could also enhance the contrast ratio. An alternative approach involves enhancing the fabrication process to achieve higher precision in structures production. By implementing advanced lithographic techniques

and optimized deposition methods, we can minimize structural imperfections that lead to the designed phase gradient fidelity variations. This includes reducing material porosity and other fabrication-induced defects.

We have added the discussions of the experimental contrasts in the revised manuscript and the simulation results in the supplementary materials.

Comments 5. *The meta-LCoS devices were fabricated using electron beam lithography overlay. How much overlay errors are acceptable without affecting the device performance?*

Response. We thank the reviewer for the comments.

To quantitatively assess the impact of overlay errors on device performance, we have simulated the total efficiency of the device (where total efficiency is equal to the product of the diffraction efficiency and reflectivity) across a range of overlay errors (0-100 nm). As illustrated in Fig. R1.5, the efficiency decreases gradually with the increase of overlay errors and degrades particularly sharply beyond the 90 nm. This abrupt transition indicates that the device is beginning to fail. Consequently, we have set 90 nm as the upper tolerance limit for overlay accuracy in the fabrication process.

We have added the discussions and the simulation results in the supplementary materials to ensure the completeness of the manuscript.

Fig. R1.5. The effect of overlay errors on device efficiency. a, Schematic diagram of overlayer alignment error in a supercell. **b,** Simulated total efficiency of the off-axis reflection light at different overlayer error with operational wavelength of 532 nm.

Comments 6. For practical applications like projectors or AR/VR displays that demand high resolutions (e.g., 1080p or 4K), what challenges arise in scaling the meta-LCoS technology from its current prototypes (64-pixel monochrome and 9-pixel color)? How are issues such as pixel crosstalk, fabrication yield, and maintaining contrast at smaller pixel sizes addressed? Does the metasurface layer introduce unique thermal management challenges, and if so, what strategies could mitigate them?

Response. We thank the reviewer for this comprehensive inquiry into the scalability of our technology.

Actually, in order to extend to higher resolutions devices, our research has already taken into account some feasible factors in practical applications. The meta-LCoS device is designed as a structure consisting of Al metasurface, SiO₂ and Al backplane to simulate the composition of a traditional LCoS panel structure and be compatible with existing manufacturing processes. This ensures that the technology can leverage established manufacturing infrastructure, and allows for the realization of complex functionalities

without major overhauls of the production line. Moreover, to protect the panel pads during fabrication, several measures are taken. These include the use of protective coatings and the careful design of the pad layout to minimize exposure to damaging agents. Additionally, the choice of materials and the optimization of each fabrication step are crucial to prevent any damage to the panel that could compromise the electrical connections and the overall performance of the device.

However, as reviewer mentioned, several key challenges still arise when scaling meta-LCoS technology to meet the demands of high-resolution applications like projectors or AR/VR displays. First of all, as pixel sizes shrink, the risk of optical and electrical crosstalk between pixels increases. Light leakage between pixels can further compromise contrast. This can lead to inaccurate pixel modulation and degrade image quality. Additionally, achieving high fabrication yield becomes more difficult with smaller pixel sizes. Manufacturing defects or variations can significantly impact device performance and yield. The compatible integration process between metasurfaces and LCoS panels is also an important technical challenge that needs to be addressed. It is necessary to ensure that the pads and structure of the LCoS panel are not damaged during the manufacturing process of the metasurfaces.

Pixel crosstalk mitigation can be achieved through the etching of deep trenches between pixel electrodes or the construction of insulating barriers, such as photoresist or inorganic materials, to impede the lateral diffusion of electric fields (reference DOI:10.1143/JJAP.46.2454). Furthermore, the employment of voltage compensation algorithms enables the dynamic fine-tuning of the drive voltage, contingent on the status

of adjacent pixels, thereby mitigating the effects of crosstalk (reference: LCOS spatial light modulators: trends and applications, Optical Imaging and Metrology, Advanced Technologies, 2012: 1-29 and DOI:10.1889/1.1832024). In regard to the enhancement of fabrication yield, the development of meta-atoms exhibiting reduced sensitivity to fabrication variations is imperative. This objective can be accomplished through meticulous design and simulation, thereby ensuring that the meta-atoms maintain functionality despite geometric deviations. Furthermore, the employment of advanced CMOS processes and liquid crystal packaging methods is also a viable option. Maintaining adequate contrast necessitates the mitigation of light leakage in the “off” state. The pixel structure must be improved to minimize light leakage between pixels. Furthermore, it is imperative to meticulously select the materials for the LC and nanograting, ensuring that their refractive indices are compatible.

For thermal management challenges, as can be seen from the Fig. R1.6, the electric field coupling of our device is mainly limited to the Al nanorods, but the coupling strength is small and only produces a weak thermal effect. Currently, LCoS chip heat dissipation is achieved through active water cooling (especially in kilowatt-level laser processing scenarios), passive heat conduction materials (carbon fiber, graphene composite materials), and structural design (multi-layer interface optimization) for thermal management. These are all relatively mature technologies. Future designs in our device could incorporate additional heat dissipation features, such as heat sinks or thermal interface materials, to further improve heat management and ensure the device operates within safe thermal limits. In the supplementary materials, we have also proposed a

scheme that utilizes dielectric nanopillars to achieve polarization-insensitive amplitude modulation. This scheme leverages the propagation phase and geometric phase of the nanopillars. In practical applications, the optimized dielectric nanopillars can be used to avoid thermal generation in the structure.

We have added a discussion on high-resolution practical applications and thermal management on the revised manuscript to strengthen the manuscript's relevance for practical device deployment.

Fig. R1.6 Distribution of electric fields of the super unit cell at “off” and “on” state.

Comments 7. *Switching times of 40 ms and 56 ms are reported. What factors (e.g., LC viscosity, layer thickness, anchoring strength) most influence these times, and what strategies (e.g., thinner LC layers, alternative materials) could reduce them to below 10 ms?*

Response. We thank the reviewer for the comments.

For a LC device, the total switching time consists of the sum of the fall time and rise time, which respectively represent the time required for LC molecules to rotate to a certain angle under an electric field and the relaxation time required for the LC to return to its initial position after the external excitation ceases. The rise time t_r and fall time t_f of a LC device can be expressed as (reference DOI:10.1364/AO.28.000048):

$$t_r = t_0 = \frac{\gamma_1 d^2}{K_{11} \pi^2}$$
$$t_f = \frac{t_0}{\left| \left(\frac{V}{V_{th}} \right)^2 - 1 \right|}$$

where t_0 represents the relaxation time of the LC, γ_1 is the rotational viscosity of the LC, K_{11} is the bending elasticity coefficient, V is the applied voltage, and V_{th} is the threshold voltage of the LC, which is influenced by the anchoring strength.

Selecting LC materials with lower viscosity can enhance the switching speed. Low viscosity materials allow for faster reorientation of LC molecules under the same electric field conditions. In the configuration under our consideration, only the LC, positioned at the same height as the nanograting, contributes to the amplitude modulation of the device. Consequently, both the anchoring force of the nanograting and the voltage sharing of the metasurface structure have a significant impact on the switching time. From the equation,

it can be seen that to improve the switching speed, one of the most effective ways to decrease switching time is to reduce the thickness of the LC layer. As the switching time is proportional to the square of the LC thickness, even a small reduction in thickness can lead to a significant decrease in switching time. The current device configuration employs a 3 μm LC layer, yielding switching times of 40 ms and 56 ms. Therefore, reducing the LC thickness represents the most direct and effective approach to achieve sub-10 ms switching.

We have added the discussion of methods for increasing switching speed in the revised manuscript.

Comments 8. *For the monolithic color meta-LCoS, how does the color gamut and accuracy compare to traditional LCoS systems? Please provide quantitative metrics (e.g., CIE coordinates) or discuss crosstalk improvement beyond spatial filtering.*

Response. We thank the reviewer for the comments.

The available color base of traditional LCoS projection equipment is determined by the spectral range of light emitted by the light source. Different light sources vary in terms of color gamut and accuracy. In LCoS projection equipment, the beam splitter and beam combining system separate the light from the light source into the three primary colors of red, green, and blue. Then, they recombine the modulated colors to form a color image. The optical performance of these systems, including the beam splitting ratio and transmittance, affects color accuracy and the range of the color gamut. Poor performance of these systems may result in uneven color mixing and a narrower color gamut. For LCoS projection devices that use sequential color methods, the precision of the sequential

control significantly impacts the color gamut and accuracy.

Our meta-LCoS system avoids color crosstalk caused by the beam splitter and beam combining components because it does not contain these components itself. The color gamut and accuracy of our meta-LCoS devices depend on the spectral response and dispersion of the cells. The bandwidth of the resonance determines the color purity and the efficiency of light utilization. A narrower bandwidth results in higher color purity but may also reduce light utilization efficiency due to the narrower acceptance of wavelengths. The dispersion characteristics of the metasurface also have a significant impact on color purity, because spatial filtering cannot separate light beams with similar wavelengths (if the incident wavelengths are similar, the resulting off-axis angles will also be similar). Consequently, the utilization of color filters to achieve monolithic color meta-LCoS has also been contemplated, as shown in Fig. R1.7. Narrowband filters are aligned with the subpixels of the metasurface to separate incident white light or colored light. The periods of nanorods on the subpixels are precisely engineered to ensure a uniform off-axis projection angle for all three colors. The utilization of filters has been demonstrated to substantially enhance the color gamut and precision, with numerous examples of filter applications in the display domain available for reference.

Fig. R1.7 Schematic of the monolithic color meta-LCoS projection display implemented by utilizing color filter.

We have added relevant discussion on methods for achieving color projection in the revised manuscript.

Comments 9. *Supplementary Section 8 explores dielectric nanopillars but decides for aluminum due to fabrication challenges. What specific improvements (e.g., material choice, lattice design) could make them viable?*

Response. We thank the reviewer for the comments.

Dielectric nanopillars, particularly those employing high-index materials such as silicon nitride (Si_3N_4) and titanium dioxide (TiO_2), offer superior optical performance—including enhanced polarization conversion efficiency and precise phase modulation—while maintaining low propagation losses in the visible spectrum. However, metal structures remain advantageous for proof-of-concept validation due to their compatibility with established nanofabrication techniques, such as electron-beam lithography and evaporation. The processing challenges highlighted in the manuscript primarily stem

from the inherent trade-offs between dielectric systems (optimized for mass production) and metallic systems (preferred for rapid laboratory prototyping).

Dielectric systems can be transitioned to mass production through two key strategies:

1. Material and fabrication optimization

For mass production of dielectric metasurfaces, CMOS compatible materials such as silicon nitride (Si_3N_4) and silicon carbide (SiC) should be prioritized. These materials established deposition protocols via plasma-enhanced chemical vapor deposition (PECVD) and atomic layer deposition (ALD). High resolution, high throughput and high aspect ratio nanopillars can be achieved through deep UV lithography, combined with reactive ion etching using optimized gas mixtures. Subsequent planarization via spin-on-glass refill and chemical-mechanical polishing (CMP) ensures surface uniformity for multilayer integration (reference DOI:10.1515/nanoph-2020-0063, 10.1364/OE.26.001573).

2. Design for manufacturability

Further optimizing the lattice design of the metasurface can improve the feasibility of dielectric nanopillars. Non-local design or multi-objective inverse design using intelligent algorithms can be demonstrated to adjust parameters such as the period, shape, and size of the nanopillars. This facilitates the creation of metasurface lattices with excellent optical performance and low manufacturing difficulty. (reference DOI:10.1515/nanoph-2020-0063, 10.1038/s41377-019-0159-5)

We have added a discussion of dielectric systems improvement methods as a subsequent mass production option in the supplementary material.

Reviewer 2:

Comments :

The authors demonstrate a monolithic color meta-LCoS prototype that integrates dual-layer metasurfaces to achieve polarization-insensitive, full-color amplitude modulation. The pixelated monochrome and color prototype capable of dynamically projecting diverse patterns under unpolarized illumination can be achieved by forming dual-layer subwavelength gratings composed of metal and dielectric nanostructures on pixelated LCoS backplane. The work demonstrates significant advancements in simplifying optical engines for projection and AR/VR displays. I thus recommend that this manuscript be accepted for publication in Nature Communications. The following comments and issues, once properly addressed, should help improve the manuscript.

Response. We thank the reviewer for the evaluation of our work. We appreciate the detailed comments. We are happy to receive such excellent advices for improving the manuscript.

Comments 1. *In contrast to DMD and conventional LCoS schemes, the authors mentioned that polarization-insensitive meta-LCoS can achieve simple projection displays system with unpolarized source, high contrast and high resolution. I agree with their motivations for studying polarization-insensitive meta-LCoS, but total efficiency and CMOS compatibility for integration of metasurface should also be carefully considered. More comments and discussions are suggested to strengthen the manuscript and make it more convincing and impactful.*

Response. We thank the reviewer for the comments.

As the reviewer stated, our device enables the projection display with unpolarized source, high contrast and high resolution, more importantly, simplifies the system's complexity and size. We appreciate the reviewers' insightful comments regarding the overall efficiency and fabrication compatibility of our meta-LCoS device, which indeed warrant thorough discussion. The total device efficiency is fundamentally determined by three key factors: reflectivity, polarization conversion efficiency and diffraction efficiency. In our current design, we have optimized these parameters through careful engineering of the metasurface geometry and LC alignment to maximize performance within the existing material system.

However, we acknowledge that the current efficiency still has significant room for improvement, primarily due to three factors: (1) optical losses in the metallic nanostructures, which could be mitigated by transitioning to low-loss dielectric materials; (2) limited phase control freedom in the resonant structures, potentially addressable through alternative phase modulation approaches, such as the propagation phase and geometric phase of high aspect ratio dielectric structures; and (3) cumulative fabrication tolerances across multiple processing steps, which may be improved with advanced lithography and etching techniques. Through these optimizations, we anticipate achieving total efficiencies exceeding 50%, surpassing conventional performance benchmarks.

Regarding fabrication compatibility, our design has incorporated several key features to maintain alignment with traditional LCoS manufacturing processes, including the use of standard backplane materials (Al and SiO₂) CMOS-compatible fabrication sequences. To ensure successful mass production, we are further developing solutions

such as process simplification and design rule optimization, exemplified by our work on dielectric nanopillar integration in Section 8 of the supplementary materials that maintains compatibility with existing semiconductor fabrication lines while improving optical performance. This ensures that the technology can leverage established manufacturing infrastructure, and allows for the realization of complex functionalities without major overhauls of the production line. This compatibility is crucial for the large-scale production and commercialization of the meta-LCoS devices.

This comprehensive approach to both efficiency optimization and fabrication compatibility demonstrates the strong potential of our meta-LCoS technology for practical implementation and commercialization.

We have added the discussions on enhancing overall efficiency and fabrication compatibility in the revised manuscript to make it more convincing and impactful.

Comments 2. *For the design of meta-LCoS shown in Fig. 2, I wonder why the authors chose Al metasurface, SiO₂ and Al electrode to form the reflective HWPs and subwavelength gratings. The selection of 4 - level phase and ordinary PCR (nearly 75%) in Fig. 2d results in low total efficiency (nearly 10%) in the targeted diffractive order shown in Fig. 2h. Besides, please supply the clear definition of contrast ratio.*

Response. We thank the reviewer for the comments.

1. Reasons for selecting structural materials

From design aspect, first, SiO₂ and Al electrode to simulate the composition of a traditional LCoS panel structure and be compatible with existing manufacturing processes. Second, in the visible light spectrum, Al has lower absorption and loss and better stability

than other metals such as gold and silver, which is the main reason for choosing Al as the metasurface material. At the same time, designing the metasurface unit as an Al-SiO₂-Al MIM multilayer structure can achieve phase control of surface plasmon resonance on the nanorod surface. From fabrication aspect, Al metasurfaces remain advantageous for proof-of-concept validation due to their compatibility with established nanofabrication techniques, such as electron-beam lithography and evaporation, making them a good choice for verifying the feasibility of this method. Of course, for mass production of dielectric metasurfaces, CMOS compatible materials such as silicon nitride (Si₃N₄) and silicon carbide (SiC) should be prioritized, as discussed in the previous question. These materials established deposition protocols via plasma-enhanced chemical vapor deposition (PECVD) and atomic layer deposition (ALD).

2. The discussion about efficiency

We think the 10% efficiency mentioned by the reviewer may be calculated using the normalized maximum value of the target diffraction order. Actually, the total efficiency of diffraction orders can be calculated as the ratio of the optical intensity contained within three times the full width at half maximum of the diffraction image to the total optical intensity of the incident light. The total efficiency calculated using this method is approximately 25%, as shown in Fig. R2.1. This efficiency is not particularly outstanding, of course, and may be due to the following factors. First, the overall efficiency of the target diffraction orders is influenced by a variety of factors, including the accuracy of the phase gradient, the PCR and the reflectivity of the nanorods. When selecting a nanorod, it is important to consider the phase response, PCR, and reflectance. In our

design, the selected nanorod's phase response differs from the ideal phase, resulting in decreased diffraction efficiency. Additionally, mutual coupling between nanorods in the supercell and fabrication error causes changes in the phase gradient, resulting in a further decrease in diffraction efficiency. Non-local design or multi-objective inverse design using intelligent algorithms can be demonstrated to adjust parameters such as the period, shape, and size of the nanopillars. This facilitates the creation of metasurface lattices with improved efficiency. The implementation of low-loss dielectric nanopillars has also the potential to enhance the overall efficiency of the system, as discussed in Section 8 of the supplementary materials.

Fig. R2.1 The simulation efficiency curve at the operational wavelength of 532 nm.

3. The definition of contrast ratio

Regarding the definition of contrast ratio, we apologize for not providing a corresponding definition in our previous manuscript. The contrast is defined as the ratio of the intensity of the “on” state to the intensity of the “off” state of off-axis reflected

light. In our experiments, it was measured using an optical power meter to quantify the intensity of the off-axis reflection point in both states.

We have modified the vertical axis (changing Intensity to Efficiency) of Fig. 2f and added the definition of contrast ratio in the revised manuscript for improved clarity.

Comments 3. *The design choice of a 30.8° off-axis reflection angle in the polarization-insensitive meta-LCoS (Fig. 2) requires clarification: (i) What specific projection system requirements necessitate this angular deviation from conventional on-axis configurations? (ii) How does this parameter affect the optical efficiency and contrast ratio within the targeted diffraction order? (iii) Could the implemented angular offset potentially compromise the device's performance metrics as shown in Fig. 2h?*

Response. We thank the reviewer for the comments.

1. The reason for 30.8° off-axis reflection angle

The 30.8° off-axis reflection angle was not chosen to meet specific requirements of projection systems. This angle is determined by the lattice period of the selected unit cell and is not specifically designed. For example, the period designed in the manuscript is 260 nm, which constitutes a 4-order phase profile, so the off-axis angle α is calculated as $\alpha = \sin^{-1}(\lambda/4P)$, where P is the lattice period of single unit cell and λ is the wavelength of the incident light. Of course, in the design of monolithic full-color devices, the off-axis angles of different wavelengths have been precisely engineered to ensure a uniform off-axis projection angle for all three colors to achieve three-color mixing. Specifically, the red, green, and blue subpixels achieve a consistent 30.4° off-axis projection angle with nanorods array periods of 314 nm, 264 nm, and 230 nm, respectively.

2. The influence of the off-axis angle

The off-axis optical configuration in our design effectively separates the zero-order (unmodulated) light from the modulated signal path through careful angular control. This spatial filtering mechanism significantly reduces zero-order interference, thereby enhancing image contrast and overall projection quality. The off-axis angle is related to the nanorods' lattice period, which directly impacts the metasurface's diffraction efficiency and reflectance, thereby affecting the device's overall efficiency. Thus, the lattice period must be carefully optimized to maximize optical efficiency. In the design of monolithic full-color devices, we have optimized the off-axis angle to achieve an optimal balance between optical efficiency and color system integration requirements through comprehensive numerical simulations.

3. Discussion on the impact of off-axis angle on performance indicators

As discussed above, the off-axis angle is determined by the lattice period, which is chosen to maximize efficiency. This angle naturally forms under the lattice and does not affect the efficiency shown in Fig. 2h. In addition, the off-axis optical configuration separates the zero-order light from the modulated signal path. This spatial filtering mechanism significantly reduces zero-order interference, thereby enhancing contrast shown in Fig. 2h.

The description of the off-axis in the manuscript have been updated for improved clarity and rigor.

Comments 4. *For the fabrication process of the polarization-insensitive meta-LCoS shown in Fig. 3a, I wonder whether the "Spinning HSQ" process is reliable, because the*

baked temperature of HSQ is 250 °C. It would be better to address detection protocols for aluminum oxidation in both the electrode and metasurface components. Oxidation-induced defects are likely to degrade the device's electrical conductivity and optical performance, as evidenced by potential impacts on diffraction efficiency and contrast ratio metrics.

Response. We thank the reviewer for the comments.

The “Spinning HSQ” process is employed to flatten the surface of the metasurface, thereby preparing it for the subsequent overlaying of nanograting patterning. The baking temperature of 250°C is necessary to remove solvents and solidify the HSQ layer. However, we understand your concerns about potential device defects at this temperature. During the baking process, all materials (Al, SiO₂ and HSQ) in the device are capable of withstanding this temperature. It must be acknowledged that there exist alternative methodologies, including the utilization of low-temperature curing SOG (spin-on glass) materials and the industry-standard method of filling the dielectric materials and then mechanically polishing it.

Al readily forms a thin, dense protective oxide layer in the air. This oxide layer functions as a barrier, hindering the further contact of oxygen with the aluminum interior and thereby decelerating the oxidation process. At room temperature, the maximum thickness of the oxide film is approximately 2 to 3 nm within one day (reference DOI: 10.1007/s11085-012-9299-1). Additionally, this oxide layer possesses a high melting point, thereby enabling aluminum to withstand elevated temperatures to a certain extent. During the manufacturing process, we coated the aluminum electrodes and nanorods by

sputtering SiO₂ and spinning HSQ, respectively, to prevent further oxidation. We have also simulated the impact of oxidation on device performance, as shown in Fig.R2.2. As the thickness of the oxide layer gradually increases from 0 to 20 nm, the efficiency of the device gradually decreases, but the decrease is not significant.

Fig. R2.2. The effect of oxide layer thickness on device efficiency

Comments 5. *For manufacturing tolerances, I've noticed that nanorod dimensions (e.g., ± 5 nm variations) and PMMA grating profiles are likely to influence device performance. Has a sensitivity analysis been performed to establish acceptable fabrication tolerances?*

Response. We thank the reviewer for the comments.

To systematically evaluate the fabrication tolerance of our meta-LCoS device, we performed comprehensive numerical simulations at the operational wavelength of 532 nm, examining the impact of nanorods at different processing errors and gratings at different size profiles on optical efficiency.

As illustrated in Fig. R2.3, the device maintains stable performance when

dimensional errors of nanorods remain within -20 nm of the designed dimensions. When dimensional errors exceed +5 nm, we observe a particularly sharp decline in device efficiency. This behavior can be attributed to the resonant nature of the metasurface elements. Within the -20 nm tolerance range, the nanostructures remain within their designed resonant regime, preserving both the polarization conversion efficiency and phase modulation accuracy. However, when nanorods dimensions deviate beyond this range, the meta-atoms shift out of their optimal resonance condition, leading to degraded polarization conversion and disrupted phase gradient profile, which collectively account for the observed efficiency reduction. These findings establish clear fabrication tolerance limits while demonstrating the robustness of our design to minor process variations characteristic of high-volume semiconductor manufacturing.

Fig. R2.3 The effect of dimensional errors of nanorods on device efficiency. a, Schematic diagram of dimensional error. L and W are designed dimension, while L_f and W_f are actual fabrication dimension. **b,** Simulated total efficiency of the off-axis reflection light at different dimensions error with operational wavelength of 532 nm.

To evaluate the fabrication tolerance of nanograting, we conducted numerical

simulations to analyze the relationship between grating dimensional variations and optical efficiency. As illustrated in Fig. R2.4, the device maintains stable performance across a wide range of grating size errors (± 40 nm), with efficiency fluctuations remaining within small value of the nominal value. This remarkable stability suggests that our design exhibits substantial robustness against typical fabrication variations encountered in nanoscale manufacturing processes.

The observed insensitivity to dimensional variations can be attributed to the non-resonant operation principle of the grating, where slight deviations from ideal dimensions do not significantly alter the phase modulation characteristics. These simulation results confirm that our device architecture can accommodate the inherent process variations of standard semiconductor fabrication techniques while maintaining consistent optical performance, a critical requirement for practical implementation and mass production.

Fig. R2.4 The effect of dimensional errors of nanograting on device efficiency. **a**, Schematic diagram of dimensional errors of nanograting. D is designed dimension and D_f is actual fabrication dimension. **b**, Simulated total efficiency of the off-axis reflection light at different dimensions error with operational wavelength of 532 nm.

We have added the discussions in the revised manuscript and the simulation results in the supplementary materials.

Comments 6. *For the characterization of the polarization-insensitive meta-LCoS shown in Figs. 3e-3g, the authors explained that the lower contrast ratio compared to simulated results due to inherent fabrication process and refractive index mismatch between PMMA nanograting and ordinary refractive index of LC. Have the authors considered the oxidation of Al electrode and Al metasurface? Please also discuss mitigation strategies.*

Response. We thank the reviewer for the comments.

The experimentally measured contrast ratios at operational wavelengths exhibit deviations from simulated predictions, primarily attributable to refractive index discrepancies of PMMA nanograting and fabrication-induced structural imperfections. For the oxidation of Al electrode and Al metasurface, as discussed earlier, oxidation of aluminum electrodes and metasurfaces may have little effect on device contrast due to the thin oxide layer, and further oxidation of the structure can be prevented by effectively coating the structure.

The first reason for this discrepancy is that the refractive index of the manufactured PMMA nanograting differ from the design values. In contrast, the refractive index of the two materials is set to be equal in the simulation. This mismatch in the experimental device results in incomplete destructive interference in the “off” state. As illustrated in Fig. R2.5a (Fig.4 in the reference, DOI: 10.1364/AO.54.00F139), the refractive index of PMMA ranges between 1.49 and 1.51 across the visible region. However, the fabricated PMMA nanograting exhibits a reduced effective refractive index (approximately 1.48 at

$\lambda = 532$ nm, as quantified through spectroscopic ellipsometry in Fig. R2.5b) due to porosity-induced structural variations.

Fig. R2.5 The refractive index of PMMA used in the simulation (a) and in the experiment (b).

To quantitatively assess the impact of refractive index variations on device contrast, we have performed numerical simulations with a nanograting refractive index of $n_G = 1.48$. As shown in Fig. R2.6, this configuration yields a simulated contrast ratio of 136:1, which is slightly higher than the contrast ratio achieved in the experiment.

Fig. R2.6. Numerical contrast ratio with the PMMA refractive index $n_G = 1.48$ at the operating wavelength of 532 nm.

Furthermore, fabrication and packaging imperfections contribute significantly to contrast ratio degradation. Structural deviations—including dimensional inaccuracies, morphological defects, and sidewall steepness variations—compromise the designed

phase gradient fidelity, thereby attenuating destructive interference in the “off” state. During encapsulation, incomplete LC infiltration induces microbubble formation, further distorting the phase gradient of the device. These combined effects reduce the maximum achievable contrast compared to ideal simulated conditions.

To mitigate these effects and improve the contrast ratio, we propose the following solutions. To address the refractive index mismatch between PMMA and the n_o of the LC, strategic doping of high-index nanoparticles (e.g., TiO_2 , $n \approx 2.5$ at 532 nm) into the PMMA presents a viable solution (reference DOI: 10.1002/app.32567, 10.1016/j.ijleo.2014.04.077). The use of another nanograting materials with refractive index that more closely match the n_o of the LC could also enhance the contrast ratio. An alternative approach involves enhancing the fabrication process to achieve higher precision in structures production. By implementing advanced lithographic techniques and optimized deposition methods, we can minimize structural imperfections that lead to the designed phase gradient fidelity variations. This includes reducing material porosity and other fabrication-induced defects.

We have added the discussions of the experimental contrasts in the revised manuscript and the simulation results in the supplementary materials.

Comments 7. *The drive voltage that a CMOS backplane can apply is generally within the range of 0 to 5V. For the experimental results shown in Figs. 3e-3g, the authors explained that the drive voltage is primarily influenced by the intrinsic characteristics of the LC and its thickness. The optimization of intrinsic material parameters (e.g., refractive index anisotropy, elastic constants) within high-birefringence liquid crystal (LC) materials*

requires systematic investigation to achieve target modulation depth through tailored electro-optic response engineering. More in-depth discussions are needed to better support the conclusions.

Response. We thank the reviewer for the comments.

A highly efficacious and expedient approach to reducing the drive voltage entails a reduction in the thickness of the LC layer. The voltage division of the liquid crystal results in a thinner liquid crystal requiring a lower voltage to achieve the same electric field. The selection of liquid chromatography materials with reduced viscosity can also result in a decrease in drive voltage. The reorientation of LC molecules is accelerated by low-viscosity materials under constant electric field conditions. We performed precise measurements of the intrinsic parameters of high birefringent LC to ensure the rigor of our work, as shown in Table R2.1.

Table R2.1 The intrinsic parameters of high birefringent LC

Clearing Point (°C)	124.4
Rotary Viscosity(mpa-s,70°C)	100.6
Δn (532 nm, 25°C)	0.42
n_o (532 nm, 25°C)	1.5
$\Delta\epsilon$ (1KHz, 70°C)	15.6
ϵ_{\perp} (1KHz, 70°C)	4.2
K_{11} (pN, 25°C)	20.1
K_{22} (pN, 25°C)	/
K_{33} (pN, 25°C)	25.6

We have added the discussions on material parameters within high-birefringence LC in the supplementary materials to make manuscript more convincing and impactful.

Comments 8. *A systematic evaluation is needed to establish device performance benchmarks through a quantitative comparison of polarization-dependent total efficiency between conventional LCoS and meta-LCoS configurations, as shown in Figs. 3h–3j, across varying incident polarization states.*

Response. We thank the reviewer for the comments.

We appreciate the reviewer's suggestion for a quantitative comparison to establish device performance benchmarks. The primary goal of our study is to demonstrate the polarization-insensitive characteristics of the meta-LCoS device. However, we understand the importance of evaluating and comparing the total efficiency in practical applications. Here, we provide a detailed response addressing the evaluation method of polarization-dependent total efficiency.

1. Evaluation method of polarization-dependent total efficiency

The total efficiency defined as the ratio of the intensity of the off-axis reflected light (i.e. the modulated light) to the intensity of the incident light. Since we are mainly concerned with polarization-insensitive properties, we compare the modulated light intensity variation of conventional LCoS and meta-LCoS devices at different polarizations. The ratio of modulated to incident light (I_{RL}/I_{IL}) is normalized to highlight this variation, as shown in the Fig. 3h-3j. The evaluation method used in our study is designed to highlight the polarization-insensitive characteristics of the meta-LCoS device.

For conventional LCoS devices, we observe strong polarization-dependent behavior

across different incident polarization angles. The efficiency decreases progressively as the polarization state deviates from the optimal alignment, ultimately reaching near-zero efficiency when the incident polarization becomes perpendicular to the LC orientation. This fundamental limitation severely restricts their application in systems with uncontrolled or varying polarization states.

In contrast, our meta-LCoS device demonstrated remarkable polarization stability, maintaining consistent off-axis reflection intensity across all polarization angles. The observed polarization insensitivity represents a significant advancement for practical applications where light source polarization cannot be predetermined or controlled.

2. Practical implications

The polarization-insensitive characteristic of the meta-LCoS device has several practical implications:

Simplified System Design: By eliminating the need for polarization control components, the meta-LCoS device simplifies the overall system design, reducing complexity and cost. Additionally, the monolithic color prototype capable of generating diverse projection patterns with high contrast using an unpolarized LED light source without any polarization elements is demonstrated. This is particularly beneficial in applications such as AR/VR displays and pico-projectors, where compactness and simplicity are crucial.

Consistent Display Quality: The meta-LCoS device ensures consistent display quality regardless of the polarization state of the incident light. This is particularly important in applications where the polarization of the light source may vary, such as in

unpolarized LED illumination.

In conclusion, while our study primarily focused on demonstrating the polarization-insensitive characteristics of the meta-LCoS device. The evaluation method used effectively highlights the polarization-insensitive nature of the meta-LCoS device, which is a significant advantage in practical applications.

We have added a discussion comparing the polarization-dependent efficiency of conventional LCoS and meta-LCoS in the revised manuscript to make it more convincing.

Comments 9. *For the SEM images of the device shown in Figs. 4b-d, the active region is at the microscale, and the HSQ spacer layer appears to have been flattened before PMMA nanograting fabrication. Given the flatness and uniformity of the spacer layer, I question whether the proposed 'Spinning HSQ' scheme is suitable for LCoS chips in centimeter-scale applications beyond 2K resolution. Could the authors address potential fabrication challenges for high-resolution meta-LCoS devices?*

Response. We thank the reviewer for the comments.

We acknowledge the valid concerns regarding the practical implementation of spinning HSQ techniques for LCoS manufacturing. The proposed “spinning HSQ” scheme is exclusively employed for the purpose of flattening experimental demonstrations. In the context of higher-resolution industrial applications, alternative CMOS-compatible processes exist, such as the industry-standard method of filling the dielectric materials and then mechanically polishing it.

Actually, in order to extend to higher resolutions devices, our research has already taken into account some feasible factors in practical applications. The meta-LCoS device

is designed as a structure consisting of Al metasurface, SiO₂ and Al backplane to simulate the composition of a traditional LCoS panel structure and be compatible with existing manufacturing processes. Moreover, to protect the panel pads during fabrication, several measures are taken. These include the use of protective coatings and the careful design of the pad layout to minimize exposure to damaging agents. Additionally, the choice of materials and the optimization of each fabrication step are crucial to prevent any damage to the panel that could compromise the electrical connections and the overall performance of the device.

However, several key challenges still arise when scaling meta-LCoS technology to meet the demands of high-resolution applications like projectors or AR/VR displays. First of all, as pixel sizes shrink, the risk of optical and electrical crosstalk between pixels increases. Light leakage between pixels can further compromise contrast. This can lead to inaccurate pixel modulation and degrade image quality. Additionally, achieving high fabrication yield becomes more difficult with smaller pixel sizes. Manufacturing defects or variations can significantly impact device performance and yield. The compatible integration process between metasurfaces and LCoS panels is also an important technical challenge that needs to be addressed. It is necessary to ensure that the pads and structure of the LCoS panel are not damaged during the manufacturing process of the metasurfaces.

Pixel crosstalk mitigation can be achieved through the etching of deep trenches between pixel electrodes or the construction of insulating barriers, such as photoresist or inorganic materials, to impede the lateral diffusion of electric fields (reference DOI:10.1143/JJAP.46.2454). Furthermore, the employment of voltage compensation

algorithms enables the dynamic fine-tuning of the drive voltage, contingent on the status of adjacent pixels, thereby mitigating the effects of crosstalk (reference: LCOS spatial light modulators: trends and applications, Optical Imaging and Metrology, Advanced Technologies, 2012: 1-29 and DOI:10.1889/1.1832024). In regard to the enhancement of fabrication yield, the development of meta-atoms exhibiting reduced sensitivity to fabrication variations is imperative. This objective can be accomplished through meticulous design and simulation, thereby ensuring that the meta-atoms maintain functionality despite geometric deviations. Furthermore, the employment of advanced CMOS processes and liquid crystal packaging methods is also a viable option. Maintaining adequate contrast necessitates the mitigation of light leakage in the “off” state. The pixel structure must be improved to minimize light leakage between pixels. Furthermore, it is imperative to meticulously select the materials for the LC and nanograting, ensuring that their refractive indices are compatible.

We have added a discussion on high-resolution practical applications on the revised manuscript to strengthen the manuscript’s relevance for practical device deployment.

Comments 10. *Regarding the dynamic projection displays driven by the polarization-insensitive meta-LCoS shown in Fig. 4i, the projection images of the 64 pixels in the “on” state appear to exhibit heterogeneity. Can the authors provide an explanation for this observation and specifically comment on its implications for high-resolution meta-LCoS devices?*

Response. We thank the reviewer for the comments.

The heterogeneity of the projection images of the 64 pixels in the "on" state is

primarily caused by the following factors:

First, despite the implementation of rigorous fabrication controls, slight variations in the dimensions of the meta-LCoS pixels' structures may occur. These variations can lead to differences in light modulation efficiency, resulting in heterogeneous projection images. Second, the alignment of LC molecules may not be perfectly uniform across all pixels. Such variations can be attributed to the presence of small bubbles and contaminants during the LC packaging process, resulting in inconsistent phase modulation and light scattering. Additionally, slight discrepancies may be observed in the drive signals applied to each pixel. It is imperative to note that variations in voltage levels or timing can affect the LC response, resulting in heterogeneous optical output.

The fabrication of high-resolution devices necessitates an even greater degree of precision. The implementation of advanced lithography and etching techniques has been demonstrated to assist in the minimization of pixel-to-pixel variations, thereby enhancing the uniformity of the resultant pattern. Additionally, ensuring homogeneous LC alignment is imperative. This objective can be accomplished through the implementation of optimized alignment layers and the execution of rigorous cleaning protocols during the fabrication process, with the aim of minimizing contaminants. Precise regulation of drive signals is imperative for the effective functioning of the system. The employment of high-precision digital controllers need be demonstrated to facilitate the uniform application of voltage across all pixels, thereby mitigating disparities in optical output.

We have added the discussion of pixel heterogeneity in the revised manuscript.

Comments 11. *The 9-pixel color prototype, while a proof of concept, is insufficient to*

demonstrate high-resolution color projection. The scalability to higher pixel densities (e.g., 8K) remains unaddressed. How is spatial filtering optimized for multi-wavelength alignment?

Response. We thank the reviewer for the comments.

The 9-pixel color LCoS prototype we have developed is indeed only a proof of concept, but this method can in principle support high resolution. This is because off-axis reflection switching depends solely on the phase gradient period of the metasurface supercell rather than pixel count. The pixel size and count can be optimized using mature CMOS processes. Of course, as discussed earlier, improving resolution still requires addressing challenges such as pixel crosstalk, yield degradation, and process compatibility, which can be resolved through advanced manufacturing methods (e.g., deep trench isolation and deep UV lithography processes).

The multi-wavelength operation of our meta-LCoS system relies on carefully engineered spatial filtering schemes where the off-axis reflection angles for different colors are precisely controlled through tailored lattice periods of the metasurface unit cells. By adjusting the nanorod periodicity specifically for each color channel, we ensure a uniform projection angle across all three primary colors, enabling effective color mixing in the far field. An alternative approach we have explored involves integrating narrowband color filters with the metasurface subpixels, as shown in Fig. R2.7. In this configuration, the filters are aligned with the subpixels of the metasurface to separate the incident broadband light while the nanorod periods are similarly optimized to maintain angular consistency for all wavelengths. This filtered approach offers potential

advantages in pixel crosstalk mitigation, drawing upon established applications of spectral filtering in high-resolution display technologies. Both methods demonstrate viable pathways toward achieving full-color operation in monolithic meta-LCoS devices, with the choice between them depending on specific performance requirements and implementation constraints.

We have added relevant discussion on methods for achieving color projection in the supplementary information.

Fig. R2.7 Schematic of the monolithic color meta-LCoS projection display implemented by utilizing color filter.

Comments 12. *For practical display applications, CMOS compatibility is crucial for meta-LCoS devices. I noticed that the authors have begun to discuss the design of dielectric nanopillars for meta-LCoS in the supplementary materials. Can the authors elaborate on potential challenges in device fabrication using the CMOS-compatible process to incorporate dielectric nanopillars?*

Response. We thank the reviewer for the comments.

1. Challenges in CMOS-compatible dielectric nanopillars fabrication

The implementation of dielectric nanostructures in CMOS-compatible processes faces two significant technical hurdles. First, material compatibility presents a primary challenge, as high-index materials require precise control of deposition and etching parameters to prevent damage to underlying CMOS circuitry. Second, yield management becomes increasingly difficult at nanoscale dimensions, where minor process variations can significantly impact device performance and uniformity across wafers. The development and implementation of these specialized processes also entail substantial initial costs for equipment modification and process qualification.

2. Solutions for CMOS-compatible dielectric nanopillars fabrication

The challenges can be effectively addressed through strategic material and process. Although we chose TiO_2 in our simulation, materials with better process compatibility, such as Si_3N_4 and SiC , can also be selected. These dielectric materials offer improved CMOS compatibility while maintaining suitable optical properties. The specific preparation process can also be further optimized. Advanced lithography solutions including deep ultraviolet lithography and nanoimprint lithography can replace electron beam lithography to achieve high-resolution, low-cost patterning at production scales. When combined with coating and etching processes, these solutions enable stable, high-yield structural processing. Currently, CMOS-processed wafer-level metasurfaces made of materials such as TiO_2 (reference DOI: 10.1021/acsphotonics.1c00609) and Si_3N_4 (reference DOI: 10.1364/OPTICA.5.000825) have been reported. Planarization techniques leveraging chemical-mechanical polishing with optimized slurries ensure

surface uniformity for subsequent processing steps. (reference DOI: 10.1364/OFC.2019.W1C.2). Finally, the preparation of nanogratings is achieved by combining photolithography and overlay processes. Therefore, the dielectric nanopillars scheme shows great potential for mass production and commercial viability.

Reviewer 3:

Comments :

The work entitled "Meta-Optics Redefines Microdisplay: Monolithic Color LCoS without Polarization Dependency" by Ou et al, introduces a novel meta-LCoS microdisplay that integrates dual-layer metasurfaces onto a conventional LCoS panel.

The approach utilizes a bottom aluminium metasurface to convert polarization, enabling liquid crystal (LC) to modulate both polarization states equally. To address the absence of polarization-dependent amplitude control, the authors incorporate an upper metasurface layer made of PMMA/LC nanogratings. This layer modifies interference of outgoing light in the lateral direction, allowing dynamic control of the phase difference between the nanograting and LC regions. This adjustment facilitates constructive and destructive interference, effectively managing the "On" and "Off" states of each pixel.

By merging metasurface-enabled polarization conversion with voltage-controlled LC phase modulation, the device eliminates the inherent polarization sensitivity of traditional LCoS systems. This innovation tackles a critical limitation of conventional LCoS displays—their reliance on linearly polarized light and the need for bulky, costly polarizing optics. The study experimentally validates the concept with a 64-pixel monochrome prototype operating at three key wavelengths (465 nm, 532 nm, and 633 nm). Additionally, a three-chip architecture optical engine prototype is proposed for color projection displays. However, the 9-pixel single-chip full-color prototype exhibits significant crosstalk, attributed to the suboptimal resonant performance of the bottom metasurface. Despite this, the proposed concept and demonstration of a monolithic color

display hold substantial significance. This work marks a notable advancement in microdisplay technology, introducing a monolithic, polarization-insensitive meta-LCoS with the potential to streamline optical systems and enhance performance in next-generation AR/VR, HUD, and projection display applications.

Therefore, I recommend publication pending revisions. Below are some points that could be addressed to enhance the manuscript's clarity:

Response. We thank the reviewer for the evaluation of our work. We appreciate the detailed comments. We are happy to receive such excellent advices for improving the manuscript.

Comments 1. *While the integration of metasurface technology with LCoS is promising, the manuscript would benefit from a clearer comparison (such as a table) with existing approaches and a detailed discussion on how this work uniquely advances the state-of-the-art in terms of performance, device complexity, and integration capabilities.*

Response. We thank the reviewer for the comments.

we have compared our device with current microdisplay chips (Table R1), highlighting its superiority:

Table R1. Parameter comparison of different microdisplay technologies.

Parameters	DMD	Conventional LCoS	Our meta-LCoS
Lighting source	unpolarized illumination	polarized illumination	unpolarized illumination
Maximum resolution	4K	8K and beyond	8K and beyond
Contrast ratio	$\sim 10^3:1$	$\sim 10^3:1$	$\sim 10^3:1$ ^[1]
Color display	three-chip or time-multiplexed monolithic	three-chip or time-multiplexed monolithic	Non-time multiplexed monolithic
Smallest pixel pitch	$>5 \mu\text{m}$	$>3 \mu\text{m}$	$<3 \mu\text{m}$ ^[2]

[1]: data from simulation; [2]: theoretical data.

By integrating dual-layer metasurfaces, our meta-LCoS technology achieves high-

contrast, polarization-insensitive optical switching, which eliminates the requirement for bulky polarizing optics and beam-splitting components used in traditional LCoS systems. This significantly reduces the overall system complexity and is a significant improvement over traditional LCoS that suffers from polarization sensitivity and limited light utilization efficiency. By embedding red, green, and blue metasurface subpixels and meticulously designed off-axis angles, enabling direct color synthesis through a unified device. This eliminates the need for sequential color projection or complex multi-chip architectures. Its single-chip architecture and compatibility with standard fabrication processes make it an attractive solution for high-resolution and high-efficiency display applications.

We have added the discussions in the revised manuscript and the table in the supplementary materials.

Comments 2. *Although the full-color prototype represents a significant step forward, the proposed single-chip color display is limited by its broad spectral response, requiring a spatial filter and leading to reduced light utilization efficiency. The current design requires a resonant metasurface, yet challenges remain with the metallic structure used. Further details on the resonance performance (such as spectral response) of the selected unit cells are necessary. Including simulation results and a discussion on the current limitations, particularly regarding crosstalk caused by metallic structures, would significantly strengthen the analysis.*

Response. We thank the reviewer for the comments.

To achieve full-color projection on a single chip, we use wavelength-specific off-

axis angle engineering. This approach use a spatial filter to mitigate crosstalk between different colors, enabling color synthesis through a unified device. However, like reviewer mentioned, the spatial filter leads to reduced light utilization efficiency as some light is inevitably filtered out. We appreciate the reviewers' suggestions and have simulated the spectral response of the selected unit cells, as shown in Fig. R3.1.

Fig. R3.1 The simulated spectral response of the selected unit cells at the three operational wavelengths.

The spectral response of the unit cells is characterized by a resonant peak at the target wavelength with a certain bandwidth. The bandwidth of the resonance determines the color purity and the efficiency of light utilization. A narrower bandwidth results in higher color purity but may also reduce light utilization efficiency due to the narrower acceptance of wavelengths. Our simulations show that the unit cells exhibit strong resonance at the target wavelengths, with high transmission efficiency within the resonance peak. The simulated spectral response indicates that the resonance peaks are well-defined, and the bandwidths are optimized to balance color purity and light utilization efficiency. Our supplementary materials also explore the use of dielectric metasurfaces, which can offer lower losses and reduced crosstalk compared to metallic structures. Dielectric materials can provide high refractive index contrast while maintaining low absorption, potentially improving the efficiency and color purity of the device.

We have added the discussions in the revised manuscript and the simulation results in the supplementary materials to enhance the comprehensiveness of the analysis.

Comments 3. *The authors state that in the "Off" state, dark pixels result from light being coupled into the evanescent wave domain. In the characterized sample, could this lead to substantial heating within the device? Might it impact LC performance or the integrity of the LC encapsulation? The authors should provide commentary on these potential effects.*

Response. We thank the reviewer for the comments.

In the "off" state, the device couples light into the evanescent wave domain through destructive interference, redistributing energy rather than absorbing it. As shown in the

electric field distribution (Fig. R3.2), the absence of strong field enhancement in either the “on” or “off” state ensures minimal additional heating from optical absorption. This means that the “off” state itself does not contribute significantly to device heating.

However, prolonged operation can still lead to gradual temperature rise due to ambient heating and residual absorption in the materials. Elevated temperatures affect LC performance by reducing its birefringence (Δn), which in turn decreases phase modulation efficiency and slightly lowers the “on” state intensity. Importantly, this does not degrade the device’s contrast ratio, as the “off” state is defined by the ordinary refractive index (n_o) of the LC, which remains stable.

While the encapsulation materials (including spacers, frame glue, orientation layer, etc.) provide sufficient thermal stability to prevent LC leakage or degradation, sustained heating could still influence long-term reliability. Mitigation strategies, such as passive heat dissipation through the device’s structural layers or active thermal management in high-power applications, may be considered for extended operation.

We have added a discussion on thermal management on the revised manuscript to strengthen the manuscript’s relevance for practical device deployment.

Fig. R3.2 Distribution of electric fields of the super unit cell at “off” and “on” state.

Comments 4. *In simulation, green and red incidences show similar maximum reflected intensity (fig 2e). However, the experimental results in fig 3f and 3g show the maximum intensity for red being nearly double that of green. What accounts for this discrepancy?*

Response. We thank the reviewer for the comments.

We apologize for misleading the reviewer by making it appear as if the image represented the maximum reflected intensity of incident light at different wavelengths. Actually, the red and green lines represent the intensity variation curves of off-axis

reflection when the refractive index of the nanograting is 1.6 and 1.7, respectively, rather than the maximum reflected intensity. Fig. R3.3 (i.e. Fig.2e in the manuscript) illustrates the analysis of how the LC material properties and nanograting parameters affect the device performance. The figure clearly shows the achievable switching performance with different LC materials, as well as varying refractive indices and heights of the nanograting. The main reason for the difference in maximum intensity between red and green light in Fig. 3f and 3g is that incident light with different powers was used during testing. Additionally, the optical efficiency of the device at the two wavelengths affects the maximum intensity.

We have modified Fig. 2e in the revised manuscript for improved clarity.

Fig. R3.3 Calculated intensity of the off-axis reflection in response to changes in the refractive index and height of the LC and nanograting.

Comments 5. The authors attribute contrast ratio difference (1505:1 in simulation and 81.3:1 in experiments) to a mismatch in the refractive index of the PMMA nanograting with the LC. Isn't the simulation also use PMMA/LC as the grating material? The same condition should be used for fair comparison. What are potential ways to improve this.

Response. We thank the reviewer for the comments.

The experimental contrast ratios achieved at different wavelengths are indeed lower than the simulated values, primarily attributable to refractive index discrepancies of PMMA nanogratings and fabrication-induced structural imperfections.

The first reason for this discrepancy is that the refractive index of the manufactured PMMA nanograting differ from the design values. In contrast, the refractive index of the two materials is set to be equal in the simulation. This mismatch in the experimental device results in incomplete destructive interference in the “off” state. As illustrated in Fig. R3.4a (Fig.4 in the reference, DOI: 10.1364/AO.54.00F139), the refractive index of PMMA ranges between 1.49 and 1.51 across the visible region. However, the fabricated PMMA nanograting exhibits a reduced effective refractive index (approximately 1.48 at $\lambda = 532$ nm, as quantified through spectroscopic ellipsometry in Fig. R3.4b) due to porosity-induced structural variations.

Fig. R3.4 The refractive index of PMMA used in the simulation (a) and in the

experiment (b).

To quantitatively assess the impact of refractive index variations on device contrast, we have performed numerical simulations with a nanograting refractive index of $n_G = 1.48$. As shown in Fig. R3.5, this configuration yields a simulated contrast ratio of 136:1, which is slightly higher than the contrast ratio achieved in the experiment.

Fig. R3.5. Numerical contrast ratio with the PMMA refractive index $n_G = 1.48$ at the operating wavelength of 532 nm.

Furthermore, fabrication and packaging imperfections contribute significantly to contrast ratio degradation. Structural deviations—including dimensional inaccuracies, morphological defects, and sidewall steepness variations—compromise the designed phase gradient fidelity, thereby attenuating destructive interference in the “off” state. During encapsulation, incomplete LC infiltration induces microbubble formation, further distorting the phase gradient of the device. These combined effects reduce the maximum achievable contrast compared to ideal simulated conditions.

To mitigate these effects and improve the contrast ratio, we propose the following solutions. To address the refractive index mismatch between PMMA and the n_o of the LC, strategic doping of high-index nanoparticles (e.g., TiO_2 , $n \approx 2.5$ at 532 nm) into the

PMMA presents a viable solution (reference DOI: 10.1002/app.32567, 10.1016/j.ijleo.2014.04.077). The use of another nanograting materials with refractive index that more closely match the n_o of the LC could also enhance the contrast ratio. An alternative approach involves enhancing the fabrication process to achieve higher precision in structures production. By implementing advanced lithographic techniques and optimized deposition methods, we can minimize structural imperfections that lead to the designed phase gradient fidelity variations. This includes reducing material porosity and other fabrication-induced defects.

We have added the discussions of the experimental contrasts in the revised manuscript and the simulation results in the supplementary materials.